# Interpretable Image Classification with Adaptive Prototype-based Vision Transformers

**Chiyu Ma**
Dartmouth College
chiyu.ma.gr@dartmouth.edu

**Jon Donnelly**
Duke University
jon.donnelly@duke.edu

**Wenjun Liu**
Dartmouth College
wenjun.liu.gr@dartmouth.edu

**Soroush Vosoughi**
Dartmouth College
soroush.vosoughi@dartmouth.edu

**Cynthia Rudin**
Duke University
cynthia@cs.duke.edu

**Chaofan Chen**
University of Maine
chaofan.chen@maine.edu

## Abstract

We present ProtoViT, a method for interpretable image classification combining deep learning and case-based reasoning. This method classifies an image by comparing it to a set of learned prototypes, providing explanations of the form "this looks like that." In our model, a prototype consists of *parts*, which can deform over irregular geometries to create a better comparison between images. Unlike existing models that rely on Convolutional Neural Network (CNN) backbones and spatially rigid prototypes, our model integrates Vision Transformer (ViT) backbones into prototype based models, while offering spatially deformed prototypes that not only accommodate geometric variations of objects but also provide coherent and clear prototypical feature representations with an adaptive number of prototypical parts. Our experiments show that our model can generally achieve higher performance than the existing prototype based models. Our comprehensive analyses ensure that the prototypes are consistent and the interpretations are faithful. Our code is available at https://github.com/Henrymachiyu/ProtoViT.

## 1 Introduction

With the expanding applications of machine learning models in critical and high-stakes domains like healthcare [4, 14, 44, 52, 51], autonomous vehicles [32], finance [47] and criminal justice [5], it has become crucial to develop models that are not only effective but also interpretable by humans. This need for clarity and accountability has led to the emergence of prototype networks. These networks combine the capabilities of deep learning with the clarity of case-based reasoning, providing understandable outcomes in fine-grained image classification tasks. Prototype networks operate by dissecting an image into informative image patches and comparing these with prototypical features established during their training phase. The model then aggregates evidence of similarities to these prototypical features to draw a final decision on classification.

Existing prototype-based models [10, 4, 13, 26, 35, 36, 46, 27, 45, 8, 7], which are **inherently interpretable** [34] (i.e., the learned prototypes are directly interpretable, and all calculations can be visibly checked), are mainly developed with convolutional neural networks (CNNs). As vision transformers (ViTs) [15, 40, 41] gain popularity and inspire extensive applications, it becomes crucial to investigate how prototype-based architectures can be integrated with vision transformer backbones. Though a few attempts to develop a prototype-based vision transformer have been made [50, 49, 22], these methods do not provide inherently interpretable explanations of the models' reasoning because they do not project the learned prototypical features to examples that actually exist in the dataset.

38th Conference on Neural Information Processing Systems (NeurIPS 2024).

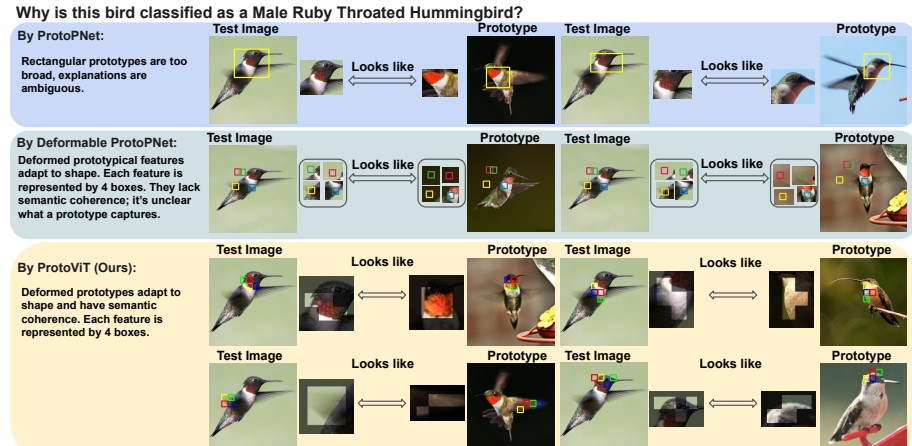

Figure 1: Reasoning process of how prototype based models identify a test image of a male Ruby-Throated Hummingbird as the correct class. Prototypes are shown in the bounding boxes.

Table 1: Table of contributions of ProtoVit (Ours) compared to existing works.

| Model | Support ViT Backbone? | Deformable Prototypes? | Coherent Prototypes? | Adaptive Sizes? | Inherently Interpretable? |
|---|---|---|---|---|---|
| ProtoPNet [10] | Yes | No | Maybe | No | Yes |
| Deformable ProtoPNet [13] | No | Yes | No | No | Yes |
| ProtoPformer [50] | Yes | No | Maybe | No | No |
| **ProtoViT (Ours)** | **Yes** | **Yes** | **Yes** | **Yes** | **Yes** |

Thus, the **cases** that these models use in their reasoning have no well defined visual representation, making it impossible to know what exactly each prototype represents.

The coherence and clarity of the learned prototypes are important. Most prototype-based models, such as ProtoPNet [10], TesNet [46] and ProtoConcept [26], use spatially rigid prototypical features (such as a rectangle), which cannot account for geometric variations of an object. Such rigid prototypical features can be ambiguous as they may contain multiple features in one bounding box (see top row of Fig. 1 for an example). Deformable ProtoPNet [13] deforms the prototypes into multiple pieces to adjust for the geometric transformation. However, Deformable ProtoPNet utilizes deformable convolution layers that rely on a continuous latent space to derive fractional offsets for prototypical parts to move around, and thus are not suitable for models like ViTs which output discrete tokens as latent representations. Additionally, the prototypes learned by Deformable ProtoPNet tend to be incoherent (see Fig. 1, middle row).

Addressing the gaps in current prototype-based methods (see Table 1), we propose the prototype-based vision transformer (ProtoViT), a novel architecture that incorporates a **ViT backbone** and can **adaptively** learn **inherently interpretable** and **geometrically variable** prototypes of **different sizes**, without requiring explicit information about the shape or size of the prototypes. We do so using a novel greedy matching algorithm that incorporates an adjacency mask and an adaptive slots mechanism. We provide global and local analysis, shown in Fig. 4, that empirically confirm the **faithfulness** and **coherence** of the prototype representations from ProtoViT. We show through empirical evaluation that ProtoViT achieves state-of-the-art accuracy as well as excellent clarity and coherence.

## 2   Related Work

***Posthoc*** explanations for CNNs like activation maximization [16, 28], image perturbation [17, 20], and saliency visualizations [3, 38, 39] fall short in explaining the reasoning process of deep neural networks because their explanations are not necessarily faithful [33, 2]. In addition, numerous efforts have been made to enhance the interpretability of Vision Transformers (ViTs) by analyzing attention weights [1, 42], and other studies focus on understanding the decision-making process through gradients [18, 19], attributions [9, 53], and redundancy reduction techniques [30]. However, because of their posthoc nature, the outcomes of these techniques can be uncertain and unreliable [2, 25].

In contrast, prototype-based approaches offer a transparent prediction process through **case-based reasoning**. These models compare a small set of learned latent feature representations called prototypes (the "cases") with the latent representations of a test image to perform classification. These models are inherently interpretable because they leverage comparisons to only well-defined cases in reasoning. The original Prototypical Part Network (ProtoPNet) [10] employs class-specific prototypes, allocating a fixed number of prototypes to each class. Each prototype is trained to be similar to feature patches from images of its own class and dissimilar to patches from images of other classes. Each prototype is also a latent patch from a training image, which ensures that "cases" are well-defined in the reasoning process. The similarity score from the test image to each prototype is added as positive evidence for each class in a "scoresheet," and the class with the highest score is the predicted class. The Transparent Embedding Space Network (TesNet) [46] refines the original ProtoPNet by utilizing a cosine similarity metric to compute similarities between image patches and prototypes in a latent space. It further introduces new loss terms to encourage the orthogonality among prototype vectors of the same class and diversity between the subspaces associated with each class. Deformable ProtoPNet [13] aims to decompose the prototypical parts into smaller sub-patches and use deformable convolutions [12, 54] to capture pose variations. Other works [35, 27, 36] move away from class-specific prototypes, which reduces the number of prototypes needed. This allows prototypes to represent a similar visual concept shared in different classes. Our work is similar to these works that define "cases" as latent patch representations by projecting the trained class-specific prototypes to the closest latent patches.

As an alternative to the closest training patches, ProtoConcept [26] defines the cases as a group of visualizations in the latent spaces bounded by a prototypical ball. Although they are not projected, the visualizations of ProtoConcept are fixed to the set of training patches falling in each prototypical ball, which establishes well defined cases for each prototype. On the other hand, works such as ProtoPFormer [50] do not project learned prototypes to the closest training patches because they observed a performance degradation after projection. The degradation is likely caused by the fact that the prototypes are not sufficiently close to any of the latent patches. The design of a "global branch" that aims to learn prototypical features from class tokens also raises concerns about interpretability, as visualizing an arbitrary trained vector (class token) does not offer any semantic connection to the input image. On the other hand, work such as ViTNet [22] also lack details about how the "cases" are established, yielding concerns about the interpretability of the model. Without a mechanism like *prototype projection* [10] or *prototypical concepts* [26] to enable visualizations, prototypes are just arbitrary learned tensors in the latent space of the network, with no clear interpretations. Visualizing nearby examples alone cannot explain the model's reasoning, since the closest patch to a prototype can be arbitrarily far away without the above mechanisms. **Simply adding a prototype layer to an architecture without well defined "cases" in the reasoning process does not make the new architecture more interpretable.**

Our work is also related to INTR [31], which trains a class specific attention query and inputs it to a decoder to localize the patterns in an image with cross-attention. In contrast, our encoder-only model, through a different reasoning approach, learns patch-wise deformed features that are more explicit, semantically coherent, and provides more detail about the reasoning process than attention heatmaps [21, 48, 6] – it shows how the important pixels are used in reasoning, not just where they are.

## 3 Methods

We begin with a general introduction of the architecture followed by an in-depth exploration of its components. We then delve into our novel training methodology that encourages prototypical features to capture semantically coherent attributes from the training images. Detailed implementations on specific datasets are shown in the experiment sections.

### 3.1 Architecture Overview

Fig. 2 shows an overview of the ProtoVit architecture. Our model consists of a feature encoder layer $f$, which is a pretrained ViT backbone that computes a latent representation of an image; a greedy matching layer $g$, which compares the latent representation to learned prototypes to compute prototype similarity scores; and an evidence layer $h$, which aggregates prototype similarity scores into a classification using a fully connected layer. We explain each of these in detail below. Let

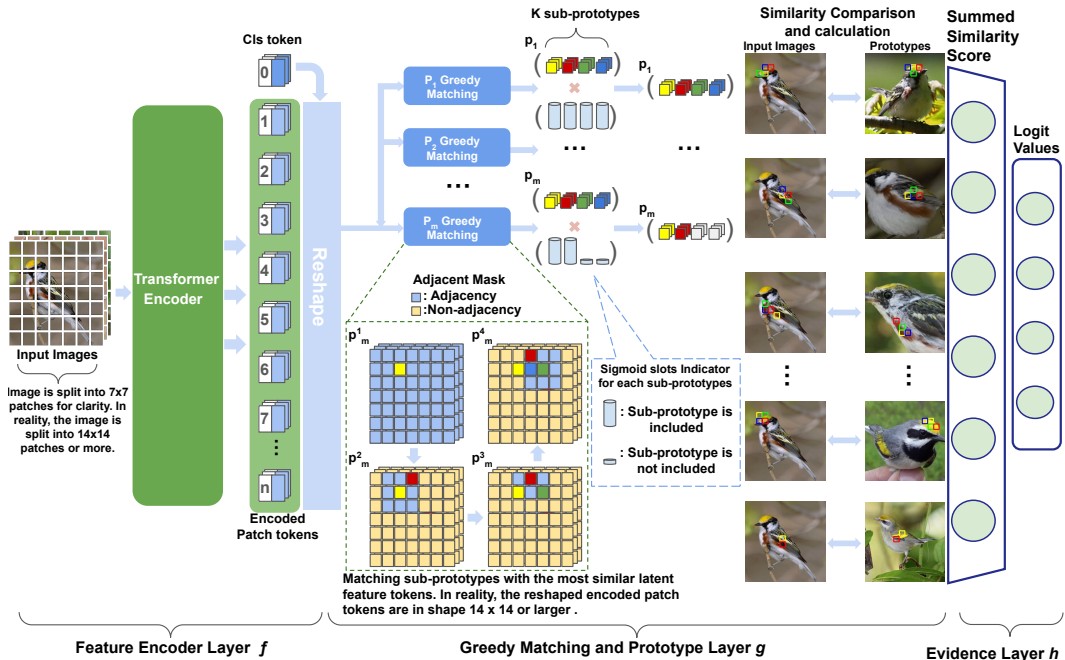

Figure 2: Architecture of ProtoViT. The feature encoder layer $f$ can be any kind of vision transformer encoders such as DeiT and CaiT. The greedy matching and prototype layer $g$ deforms the prototypes into sub-prototypes and finds the closest non-overlapping feature patches from the test image. Our adaptive slots mechanism filters out some number of sub-prototypes, and the sum of the remaining similarity scores (with a correction to avoid down-weighting prototypes with more sub-prototypes filtered out) is returned. The evidence layer $h$ is a fully connected layer that computes the logit predictions based on the summed similarity scores.

$H \times W \times C$ be the shape of an input image $\mathbf{x}$, where $H, W, C$ are the height, width and number of channels of the image respectively. The feature encoder layer $f$ is a ViT backbone, which first splits the input images into $N$ units of $L \times L \times C$ sized patches (such as the Chestnut Sided Warbler in Fig. 2), where $L$ is the height and width of the image patch. The feature encoder layer $f$ then flattens and encodes the image patches into a set of latent feature tokens $\mathbf{z}_f$ defined later in Eq. 1, where each latent feature token $\mathbf{z}_f^i \in \mathbb{R}^d$ and $d$ is the hidden dimension of the model. The network aims to learn $m$ prototypes $\mathbf{P} = \{\mathbf{p}_j\}_{j=1}^m$, where each $\mathbf{p}_j \in \mathbb{R}^{K \times d}$ is composed of $K$ sub-prototype vectors $\mathbf{p}_j^k \in \mathbb{R}^d$. The greedy matching layer $g$ then finds the most similar latent feature token $\mathbf{z}_f^i$, measured by cosine similarity, of the input image to each sub-prototype $\mathbf{p}_j^k$ without replacement (i.e., if $\mathbf{p}_j^1$ matches $\mathbf{z}_f^1$, $\mathbf{p}_j^2$ cannot match with $\mathbf{z}_f^1$). We further introduce an adjacency mask $A$ to ensure that, for each prototype $\mathbf{p}_j$, its sub-prototypes $\mathbf{p}_j^k$ are geometrically contiguous. We then introduce our adaptive slots mechanism, which decides whether each prototype $\mathbf{p}_j$ should include each sub-prototype $\mathbf{p}_j^k$ based on its coherence to the other sub-prototypes and its influence on model performance, allowing the model to learn prototypes with a dynamic number of sub-prototypes. Finally, the evidence layer $h$ computes the logit values for each class based on the summed cosine similarity score $g_{\mathbf{p}_j}^{\text{greedy}}$ for each prototype $\mathbf{p}_j$. The logit values are then normalized by a softmax function to make predictions. Intuitively, the prototypes, which are later projected to the closest latent feature tokens, can be viewed as the most representative features of each class that the model found among the training examples. The model performs classification based on the input's similarity to these key features.

## 3.2 Feature Encoder Layer

Given an input image $\mathbf{x} \in \mathbb{R}^{H \times W \times C}$, the feature encoder layer first splits the image into $N$ unit patches $\mathbf{x}_{\text{patch}}$ each of shape $L \times L \times C$. It then flattens the unit patches and projects them

to the embedding space as $\mathbf{x}_{\text{Embed}} \in \mathbb{R}^{N \times d}$, which is then prepended with a trainable class to-ken $\mathbf{x}_{\text{class}} \in \mathbb{R}^d$, and passed into the Multi-Head Attention layers along with a learnable po-sition embedding $E_{\text{pos}} \in \mathbb{R}^{(N+1) \times d}$. The ViT backbone then outputs the encoded tokens as $\mathbf{z} := [\mathbf{z}_{\text{class}}; \mathbf{z}_{\text{patch}}^1; \mathbf{z}_{\text{patch}}^2; \cdots ; \mathbf{z}_{\text{patch}}^N]$, where $\mathbf{z} \in \mathbb{R}^{(N+1) \times d}$. In this sense, each of the output patch tokens $\mathbf{z}_{\text{patch}}^i$ is the latent representation of the corresponding unit patch $\mathbf{x}_{\text{patch}}^i$. The class token can be viewed as a way to approximate the weighted sum over patch tokens, enabling an image-level repre-sentation. It is, thus, difficult to visualize the class token and therefore unsuitable for comparisons to prototypes. Drawing inspiration from the idea of focal similarity [36], which involves calculating the maximum prototype activations minus the mean activations to achieve more focused representations, we take the difference between patch-wise features and the image-level representation. In doing so, we aim to similarly produce more salient visualizations. Specifically, we define the feature token $\mathbf{z}_f$ by taking the difference between each patch token and the image-level representation. Thus, latent feature tokens can be written as:

$$\mathbf{z}_f := [\mathbf{z}_f^1; \mathbf{z}_f^2; \cdots ; \mathbf{z}_f^N], \text{ where } \mathbf{z}_f^i = \mathbf{z}_{\text{patch}}^i - \mathbf{z}_{\text{class}}, \text{ and } \mathbf{z}_f^i \in \mathbb{R}^d. \tag{1}$$

By this design, we not only encode richer semantic information within the latent feature tokens, but also enable the application of prototype layers to various ViT backbones that contains a class token. An ablation study shows that model performance drops substantially when removing the class token from the feature encoding (see Appendix Sec. E.1) .

## 3.3 Greedy Matching Layer

The greedy matching layer $g$ integrates three key components: a greedy matching algorithm, adjacency masking, and an adaptive slots mechanism, as illustrated in Fig. 2.

**Similarity Metric:** The greedy matching layer contains $m$ prototypes $\mathbf{P} = \{\mathbf{p}_j\}_{j=1}^m$, where each $\mathbf{p}_j$ consists of $K$ sub-prototypes $\mathbf{p}_j^k$ that have the same dimension as each latent feature token $\mathbf{z}_f^i$. Each sub-prototype $\mathbf{p}_j^k$ is trained to be semantically close to at least one latent feature token. To measure the closeness of the sub-prototype $\mathbf{p}_j^k$ and latent feature token $\mathbf{z}_f^i$, we use cosine similarity $\cos(\mathbf{z}_f^i, \mathbf{p}_j^k) = \frac{\mathbf{z}_f^{i\,T} \mathbf{p}_j^k}{\|\mathbf{z}_f^i\|_2 \|\mathbf{p}_j^k\|_2}$, which has range -1 to 1. The overall similarity between prototype $\mathbf{p}_j$ and the latent feature tokens $\mathbf{z}_f$ is then computed as the sum across the sub-prototypes of the cosine similarities, which has a range from $-K$ to $K$.

**Greedy Matching Algorithm:** We have not yet described how the token $\mathbf{z}_f^i$ to which $\mathbf{p}_j^k$ is compared is selected. In contrast to prior work, we do not restrict that sub-prototypes have fixed relative locations. Rather, we perform a greedy matching algorithm to identify and compare each of the $K$ sub-prototypes $\mathbf{p}_j^k$ of $\mathbf{p}_j$ to *any* $K$ non-overlapping latent feature tokens. To be exact, for a given prototype $\mathbf{p}_j$ with $K$ sub-prototypes, we iteratively identify the sub-prototype $\mathbf{p}_j^k$ and latent feature token $\mathbf{z}_f^i$ that are closest in cosine similarity, "match" $\mathbf{z}_f^i$ with $\mathbf{p}_j^k$, and remove $\mathbf{z}_f^i$ and $\mathbf{p}_j^k$ from the next iteration, until all $K$ pairs are found. This is illustrated in Fig. 2, and more details can be found in Appendix Sec. C.

**Adjacency Masking:** Without any restrictions, this greedy matching algorithm can lead to many sub-prototypes that are geometrically distant from each other. Assuming that the image patches representing the same feature are within $r \in \mathbb{R}$ spatial units of one another in horizontal, vertical and diagonal directions, we introduce the Adjacency Mask $A$ to temporarily mask out latent feature tokens that are more than $r$ positions away from a selected sub-prototype/latent feature token pair in all directions. Within each iteration $k$ of the greedy matching algorithm, the next pair can only be selected from the latent feature tokens, $\mathbf{z}_{A_j^k} = A(\{\mathbf{p}_j^{k-1}, \mathbf{z}_f^{i\,k-1}\}; \mathbf{z}_f; r)$, that are within $r$ positions of the last selected pair. An example of how the adjacency mask with $r = 1$ works is illustrated in Fig. 2. Incorporating adjacency masking with the greedy matching algorithm, we thus find $K$ non-overlapping and adjacent sub-prototype-latent patch pairs.

**Adaptive Slots Mechanism:** Because not all concepts require $K$ sub-prototypes to represent, we introduce the the adaptive slots mechanism $S$. The adaptive slots mechanism consists of learnable

vectors $\mathbf{v} \in \mathbb{R}^{m \times K}$ and a sigmoid function with a hyperparameter temperature $\tau$. We chose a sufficiently large value for $\tau$ to ensure that the sigmoid function has a steep slope. The vectors $\mathbf{v}$ are sent to the sigmoid function to approximate the indicator function as $\tilde{\mathbb{1}}_{\{\text{Include } \mathbf{p}_j^k\}} = \mathbf{Sigmoid}(\mathbf{v}_j^k, \tau)$.

Each $\tilde{\mathbb{1}}_{\{\text{Include } \mathbf{p}_j^k\}}$ is an approximation of the indicator for whether the $k$-th sub-prototype will be included in the $j$-th prototype $\mathbf{p}_j$. More details can be found in Appendix Sec. D.

**Computation of Similarity:** As described above, we measure the similarity of a selected latent-patch–sub-prototype pair using cosine similarity. We use the summed similarity across sub-prototypes to measure the overall similarity for prototype $\mathbf{p}_j$ with $K$ selected non-overlapping latent feature tokens. As pruning out some sub-prototypes for a prototype $\mathbf{p}_j$ reduces the range of the summed cosine similarity for $\mathbf{p}_j$, we rescale the summed similarities back to their original range by $K / \sum_k \tilde{\mathbb{1}}\{\text{include } \mathbf{p}_j^k\}$. We formally define the reweighted summed similarity function obtained from the greedy matching layer $g$ as:

$$g_{\mathbf{P}_j}^{\text{greedy}}(\mathbf{z}_f) = \frac{K}{\sum_{k=1}^{K} \tilde{\mathbb{1}}_{\{\text{include } \mathbf{p}_j^k\}}} \sum_{k=1}^{K} \left\{ \max_{\mathbf{z}_f^i \in \mathbf{z}_{A_j^k}} \cos(\mathbf{z}_f^i, \mathbf{p}_j^k) \right\} \cdot \tilde{\mathbb{1}}_{\{\text{include } \mathbf{p}_j^k\}}. \tag{2}$$

## 3.4 Training Algorithm

Training of ProtoViT has four stages: (1) optimizing layers before the last layer by stochastic gradient descent (SGD); (2) prototype slots pruning; (3) projecting the trained prototype vectors to the closest latent patches; (4) optimizing the last layer $h$. Note that a well-trained first stage is crucial to achieve minimal performance degradation after prototype projection. A procedure plot is illustrated in Appendix Fig. 5.

**Optimization of layers before last layer:** The first training stage aims to learn a latent space that clusters feature patches from the training set that are important for a class near semantically similar prototypes of that class. This involves solving a joint optimization problem over the network parameters via stochastic gradient descent (SGD). We initialize every slot indicator $\tilde{\mathbb{1}}_{\{\text{Include } \mathbf{p}_j^k\}}$ as 1 to allow all sub-prototypes to learn during SGD. We initialize the last layer weight similarly to ProtoPNet [10]. The slot indicators and the final layer are frozen during this training stage. The slot indicator functions are not involved in computing cluster loss or separation loss because each sub-prototype should remain semantically close to certain latent feature tokens, regardless of its inclusion in the final computation. Since we deform the prototypes, we propose modifications to the original cluster and separation loss defined in [10] to:

$$\mathcal{L}_{Clst} = -\frac{1}{n} \sum_{i=1}^{n} \max_{\mathbf{p}_j \in \mathbf{P}_{y_i}} \max_{\mathbf{p}_j^k \in \mathbf{p}_j} \max_{\mathbf{z}_f^i \in \mathbf{z}_{A_j^k}} \cos(\mathbf{z}_f^i, \mathbf{p}_j^k);$$

$$\mathcal{L}_{Sep} = \frac{1}{n} \sum_{i=1}^{n} \max_{\mathbf{p}_j \notin \mathbf{P}_{y_i}} \max_{\mathbf{p}_j^k \in \mathbf{p}_j} \max_{\mathbf{z}_f^i \in \mathbf{z}_{A_j^k}} \cos(\mathbf{z}_f^i, \mathbf{p}_j^k). \tag{3}$$

The minimization of this new cluster loss encourages each training image to have some unit latent feature token $\mathbf{z}_f^i$ that is close to at least one sub-prototypes $\mathbf{p}_j^k$ of its own class. Similarly, by minimizing the new separation loss, we encourage every latent feature token of a training image to stay away from the nearest sub-prototype from any incorrect class. This is similar to the traditional cluster and separation loss from [10] if we define the prototypes to consist of only one sub-prototype. We further introduce a novel loss term, called coherence loss, based on the intuition that, if the sub-prototypes collectively represent the same feature (e.g., the feet of a bird), these sub-prototypes should be similar under cosine similarity. The coherence loss is defined as:

$$\mathcal{L}_{Coh} = \frac{1}{m} \sum_{j=1}^{m} \max_{\mathbf{p}_j^k, \mathbf{p}_j^s \in \mathbf{P}_j; \mathbf{p}_j^s \neq \mathbf{p}_j^k} (1 - \cos(\mathbf{p}_j^k, \mathbf{p}_j^s)) \cdot \tilde{\mathbb{1}}_{\{\text{Include } \mathbf{p}_j^k\}} \tilde{\mathbb{1}}_{\{\text{Include } \mathbf{p}_j^s\}}. \tag{4}$$

Intuitively, the coherence loss penalizes the sub-prototypes that are the most dissimilar to the other sub-prototypes out of the $K$ slots for prototype $\mathbf{p}_j$. The slots indicator function is added to the

coherence loss term to prune sub-prototypes that are semantically distant from others in a prototype. Moreover, we use the orthogonality loss, introduced in previous work [46, 13], to encourage each prototype $\mathbf{p}_j$ to learn distinctive features. The orthogonality loss is defined as:

$$\mathcal{L}_{Orth} = \sum_{l=1}^{C} \|\mathbf{P}^{(l)}\mathbf{P}^{(l)^T} - \mathbf{I}_\rho\|_{\mathbf{F}}^2 \tag{5}$$

where $C$ is the number of classes. For each class $l$ with $\rho$ assigned prototypes, $\mathbf{P}^{(l)} \in \mathbb{R}^{\rho \times Kd}$ is a matrix obtained by flattened the prototypes from class $l$, and $\rho$ is the number of prototypes $\mathbf{p}_j$ for each class. $\mathbf{I}_\rho$ is an identity matrix in the shape of $\rho \times \rho$. Overall, this training stage aims to minimize the total loss as:

$$\mathcal{L}_{total} = \mathcal{L}_{CE} + \lambda_1 \mathcal{L}_{Clst} + \lambda_2 \mathcal{L}_{Sep} + \lambda_3 \mathcal{L}_{Coh} + \lambda_4 \mathcal{L}_{Orth} \tag{6}$$

where $\mathcal{L}_{CE}$ is the cross entropy loss for classification and $\lambda_1, \lambda_2, \lambda_3, \lambda_4$ are hyper-parameters.

**Slots pruning:**  In this stage, our goal is to prune the sub-prototypes $\mathbf{p}_j^k$ that are dissimilar to the other sub-prototypes for each prototype. Intuitively, we aim to remove these sub-prototypes because sub-prototypes that are not similar will lead to prototypes with an inconsistent, unintuitive semantic meaning. We freeze all of the parameters in the model, except the slot indicator vectors $\mathbf{v}$. During this stage, we jointly optimize the coherence loss defined in Eq. 4 along with the cross entropy loss. We lower the coherence loss weight to avoid removing all the slots. Since the slot indicators are approximations of step functions using sigmoid functions with a sufficiently high temperature parameter $\tau$, the indicator values during this phase are fractional but approach binary values close to 0 or 1. The loss in the stage is defined as:

$$\mathcal{L}_{\text{prune}} = \mathcal{L}_{CE} + \lambda_5 \mathcal{L}_{Coh}. \tag{7}$$

**Projection:**  In this stage, we first round the fractional indicator values to the nearest integer (either 1 or 0), and freeze the slot indicators' values. Then, we project each prototype $\mathbf{p}_j$ to the closest training image patch measured by the summed cosine similarity defined in Eq. 2, as in [10]. Because the latent feature tokens from the ViT encoder correspond to image patches, we do not need to use up-sampling techniques for visualizations, and the prototypes visualized in the bounding boxes represent the exact corresponding latent feature tokens that the prototypes are projected to.

As demonstrated in Theorem 2.1 from ProtoPNet [10], if the prototype projection results in minimal movement, the model performance is unlikely to change because the decision boundary remains largely unaffected. This is ensured by minimizing cluster and separation loss, as defined in Eq. 3. In practice, when prototypes are sufficiently well-trained and closely clustered to certain latent patches, the change in model performance after the projection step should be minimal. Additionally, this step ensures that the prototypes are inherently interpretable, as they are projected to the closest latent feature tokens, which have corresponding visualizations and provide a pixel space explanation of the "cases" in the model's reasoning process.

**Optimization of last layers:**  Similar to other existing works [10, 46], we performed a convex optimization on the last evidence layer $h$ after performing projection, while freezing all other parameters. This stage aims to introduce sparsity to the last layer weights $W$ by penalizing the $1-$norm of layer weights $\mathbf{W_{b,l}}$ (initially fixed as -0.5) associated with the class $b$ for the $l$-th class prototype $\mathbf{p}^l$, where $l \neq b$. By minimizing this loss term, the model is encouraged to use only positive evidence for final classifications. The loss term is defined as:

$$\mathcal{L}_h = \mathcal{L}_{CE} + \lambda_6 \sum_{l}^{C} \sum_{b \in \{0,...,C\}: b \neq l} \|\mathbf{W_{b,l}}\|_1. \tag{8}$$

## 4 Experiments

### 4.1 Case Study 1: Bird Species Identification

To demonstrate the effectiveness of ProtoVit, we applied it to the cropped Caltech-UCSD Birds-200-2011 (CUB 200-2011) dataset [43]. This dataset contains 5,994/5,794 images for training and testing

Table 2: Comparison of ProtoVit implemented with DeiT and CaiT backbones to other existing works. Our model is not only inherently interpretable but also superior in performance compared to other methods using the same backbone. We also include models with a CNN backbone (Densenet-161) for reference in the top section. The reported accuracies are the final results after all training stages.

| Arch. | Model | CUB Acc.[%] | Car Acc.[%] |
|---|---|---|---|
| Densenet-161 ~28.68M params | ProtoPNet (given in [10]) | 80.1 ± 0.3 | 89.5 ± 0.2 |
| | Def. ProtoPNet(2x2) (given in [13]) | 80.9 ±0.22 | 88.7 ± 0.3 |
| | ProtoPool (given in [36]) | 80.3 ± 0.3 | 90.0 ± 0.3 |
| | TesNet (given in [46]) | **81.5 ± 0.3** | **92.6 ± 0.3** |
| DeiT-Tiny ~5M params | Base (given in [50]) | 80.57 | 86.21 |
| | ViT-Net (given in [50]) | 81.98 | 88.41 |
| | ProtoPFormer (given in [50]) | 82.26 | 88.48 |
| | **ProtoViT($K$=4, $r$=1) (ours)** | **82.92 ± 0.5** | **89.02 ± 0.1** |
| DeiT-Small ~22M params | Baseline (given in [50]) | 84.28 | 90.06 |
| | ViT-Net (given in [50]) | 84.26 | 91.34 |
| | ProtoPFormer (given in [50]) | 84.85 | 90.86 |
| | **ProtoViT($K$=4, $r$=1) (ours)** | **85.37 ± 0.13** | **91.84 ± 0.3** |
| CaiT-XXS 24 ~11.9M params | Baseline (given in [50]) | 83.95 | 90.19 |
| | ViT-Net (given in [50]) | 84.51 | 91.54 |
| | ProtoPFormer (given in [50]) | 84.79 | 91.04 |
| | **ProtoViT($K$=4, $r$=1) (ours)** | **85.82 ± 0.15** | **92.40 ± 0.1** |

from 200 different bird species. We performed similar offline image augmentation to previous work [10, 46, 26] which used random rotation, skew, shear, and left-right flipping. After augmentation, the training set had roughly 1,200 images per class. We performed prototype projections on only the non-augmented training data. Additionally, we performed ablation studies on the class token (See Appendix Sec. E.1), coherence loss (See Appendix Sec. E.3), and adjacency mask (See Appendix Sec. E.2). The quantitative results for the ablation can be found in Appendix Table. 6. We assigned the algorithm to choose 10 class-specific prototypes for each of the 200 classes. Each of the prototypes is composed of 4 sub-prototypes. Each set of sub-prototypes was encouraged to learn the 'key' features for its corresponding class through only the last layer weighting, and the 4 sub-prototypes were designed to collectively represent one key feature for the class. More discussion on different choices of $K$ can be found in Appendix F. For more details about hyperparameter settings and training schedules, please refer to Appendix Sec. A and Appendix Sec. B respectively. Details and results of user studies regarding the interpretability of ProtoViT can be found in Appendix. Sec. K.

#### 4.1.1 Predictive Performance Results

The performance of our ProtoViT with CaiT and DeiT backbones is compared to that of existing work in Table 2, including results from prototype-based models using DenseNet161 (CNN) backbones. Integrating ViT (Vision Transformer) backbones with a smaller number of parameters into prototype-based algorithms significantly enhances performance compared to those using CNN (Convolutional Neural Network) backbones, particularly on more challenging datasets. Compared with other prototype-based models that utilize ViT backbones, our model produces the **highest accuracy**, **outperforming** even the black box ViTs used as backbones, while offering **interpretability**. We provide accuracy using an ImageNet pretrained Densenet-161 to match the ImageNet pretraining used in all transformer models.

#### 4.1.2 Reasoning Process and Analysis

Fig. 3 shows the reasoning process of ProtoViT for a test image of a Barn Swallow. Example visualizations for the classes with the top two highest logit scores are provided in the figure. Given this test image $\mathbf{x}$, our model compares its latent feature tokens against the learned sub-prototypes $\mathbf{p}_j^k$ through the greedy matching layer $g$. In the decision process, our model uses the patches that have the most similar latent feature tokens to the learned prototypes as evidences. In the example, our model correctly classifies a test image of a Barn Swallow and thinks the Tree Swallow is the second most likely class based on prototypical features. In addition to this example reasoning process, we conducted local and global analyses (shown in Fig. 4) to confirm the semantic consistency of prototypical features across all training and testing images, where local and global analyses are

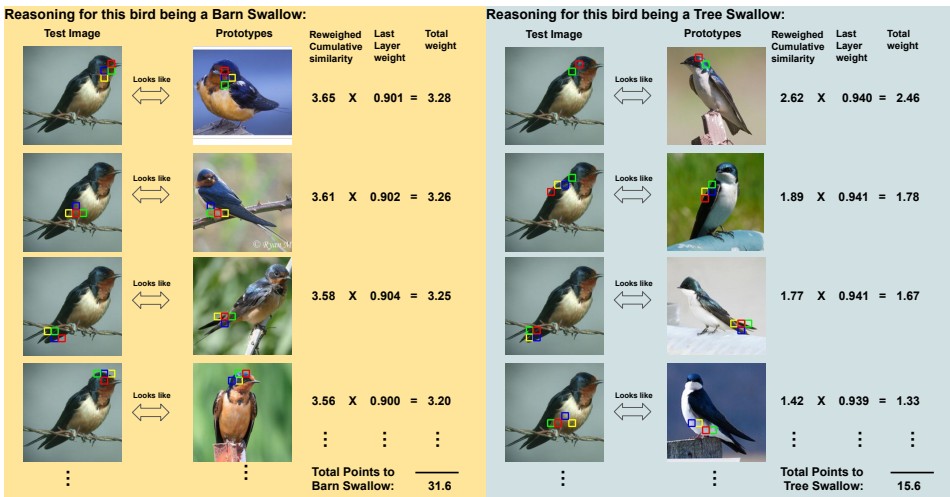

Figure 3: Reasoning process of how ProtoViT classifies a test image of Barn Swallow using the learned prototypes. Examples for the top two predicted classes are provided. We use the DeiT-Small backbone with $r$=1 and $k$=4 for the adjacency mask.

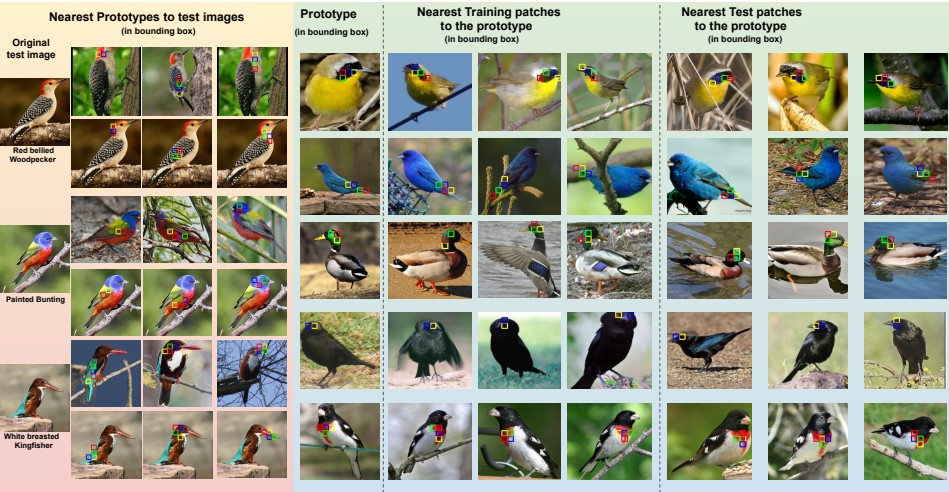

Figure 4: Nearest prototypes to test images (left), and the nearest image patches to prototypes (right). We exclude the nearest training patch, which is the prototype itself by projection.

defined as in [10]. The left side of Fig. 4 displays local analysis examples, visualizing the most semantically similar prototypes to each test image. The right side of Fig. 4 shows global analysis examples, presenting the top three nearest training and testing images to the prototypes. Our local analysis confirms that, across distinct classes and prototypes, the comparisons made are reasonable. For example, the first prototype compared to the white breasted kingfisher seems to identify the bird's blue tail, and is compared to the tail of the bird in the test image. Further, our global analysis shows that each prototype consistently activates on a single, meaningful concept. For example, the prototype in the first row of the global analysis consistently highlights the black face of the common yellowthroat at a variety of scales and poses. Taken together, these analyses show that **the prototypes of ProtoViT have strong, consistent semantic meanings**. More examples of the reasoning process and analysis can be found in Appendix Sec. H and Sec. J respectively.

### 4.1.3 Location Misalignment Analysis

Vision Transformers (ViTs) use an attention mechanism that blends information from all image patches, which may prevent the latent token at a position from corresponding to the input token at that

Table 3: Experimental results on the Location Misalignment Benchmark. We compare our ProtoViT (Deit-Small) with other prototype-based models with CNN backbones (ResNet34). We found that our model performs similarly to or better than the existing models with CNN backbones in terms of the misalignment metrics and test accuracy.

| Method | PLC | PRC | PAC | Acc. Before | Acc. After | AC. |
|---|---|---|---|---|---|---|
| ProtoPNet | $24.0 \pm 1.7$ | $13.5 \pm 3.1$ | $23.7 \pm 2.8$ | $76.4 \pm 0.2$ | $68.2 \pm 0.9$ | $8.2 \pm 1.1$ |
| TesNet | $16.0 \pm 0.0$ | $2.9 \pm 0.3$ | $3.4 \pm 0.3$ | $81.6 \pm 0.2$ | $75.8 \pm 0.5$ | $5.8 \pm 0.6$ |
| ProtoPool | $31.8 \pm 0.8$ | $4.5 \pm 0.9$ | $11.2 \pm 1.3$ | $80.8 \pm 0.2$ | $76.0 \pm 0.3$ | $4.8 \pm 0.1$ |
| ProtoTree | $27.7 \pm 0.3$ | $13.5 \pm 3.1$ | $23.7 \pm 2.8$ | $76.4 \pm 0.2$ | $68.2 \pm 0.9$ | $8.2 \pm 1.1$ |
| **ProtoViT(ours)** | $\mathbf{21.68 \pm 3.1}$ | $\mathbf{1.28 \pm 0.1}$ | $\mathbf{2.92 \pm 0.1}$ | $\mathbf{85.4 \pm 0.1}$ | $\mathbf{82.8 \pm 0.3}$ | $\mathbf{2.6 \pm 0.4}$ |

position. To assess whether ViT backbones can be interpreted as reliably as Convolutional Neural Networks (CNNs) in ProtoPNets, we conducted experiments using the gradient-based adversarial attacks described in the Location Misalignment Benchmark [37]. As summarized in Table 3, our method, which incorporates a ViT backbone, consistently matched or outperformed leading CNN-based prototype models such as ProtoPNet [10], ProtoPool [36], and ProtoTree [27] across key metrics: Percentage Change in Location (PLC), Percentage Change in Activation (PAC), and Percentage Change in Ranking (PRC), as defined in the benchmark. Lower values for these metrics indicate better performance. This shows that ProtoViT is at least as robust as CNN-based models. Moreover, as observed in the Location Misalignment Benchmark[37], the potential for information leakage also exists in deep CNNs, where the large receptive fields of deeper layers would encompass the entire image same as attention mechanisms. **While our model is not entirely immune to location misalignment, empirical results indicate that its performance is on par with other state-of-the-art CNN-based models.** Qualitative comparisons that further support this point can be found in Appendix. Sec. G, and more results and discussions on PLC for ablated models can be found in Appendix. Sec. E

## 4.2 Case Study 2: Cars Identification

In this case study, we apply our model to car identification. We trained our model on the Stanford Car dataset[24] of 196 car models. More details about the implementation can be found in Appendix. Sec. L. The performance with baseline models can be found in Table. 2. Example visualizations and analysis can be found in Appendix L. We find that **ProtoViT again produces superior accuracy and strong, semantically meaningful prototypes on the Cars dataset.**

## 5 Limitations

While we find our method can offer coherent and consistent visual explanations, it is not yet able to provide explicit textual justification to explain its reasoning process. With recent progress in large vision-language hybrid models [23, 11, 29], a technique offering explicit justification for visual explanations could be explored in future work. It is important to note that in some visual domains such as mammography or other radiology applications, features may not have natural textual descriptions – thus, explicit semantics are not yet possible, and a larger domain-specific vocabulary would first need to be developed. Moreover, as discussed in Sec. 4.1.3, our model is not completely immune to location misalignment, meaning the learned prototypical features may be difficult to visualize in local patches. However, this issue is not unique to our approach; CNN-based models face the same challenge. As layers deepen, the receptive field often expands to cover the entire image, leading to the same problem encountered with attention mechanisms in vision transformers.

## 6 Conclusion

In this work, we presented an interpretable method for image classification that incorprates ViT backbones with deformed prototypes to explain its predictions (*this* looks like *that*). Unlike previous works in prototype-based classification, our method offers spatially deformed prototypes that not only account for geometric variations of objects but also provide coherent prototypical feature representations with an adaptive number of prototypical parts. While offering inherent interpretability, our model empirically outperform the previous prototype based methods in accuracy.

# 7 Acknowledgement

We would like to acknowledge funding from the National Science Foundation under grants HRD-2222336 and OIA-2218063 and the Department of Energy under DE-SC0021358.

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

**Appendix Table of Contents**

# A Training Hyperparameters

This section documents the specific hyper-parameters used to train ProtoViT, shown in Table 4. We used identical parameter settings across all the backbones. We used the values suggested in prior work for terms that already existed, and used a small grid search to select the other values. No dedicated tuning procedure is necessary for these coefficients.

# B Training schedule

This section documents the specific training schedule used to train ProtoViT. As discussed in Sec. 3.4, the training stages are generally divided into four steps as shown in Appendix Fig. 5. Training schedule settings can be found in Appendix Table 5. Warm-up optimization is performed by training the feature encoder with a extremely small learning rate while training the prototype vectors for 5 epochs. Then, we increase the learning rate for the feature encoder layer for the following 10 epochs. As mentioned in Appendix Sec. A, the number of sub-prototypes being pruned during the slots pruning stage depend on the learning rate of the slots parameters and the weight of the coherence loss in the slots pruning stage.

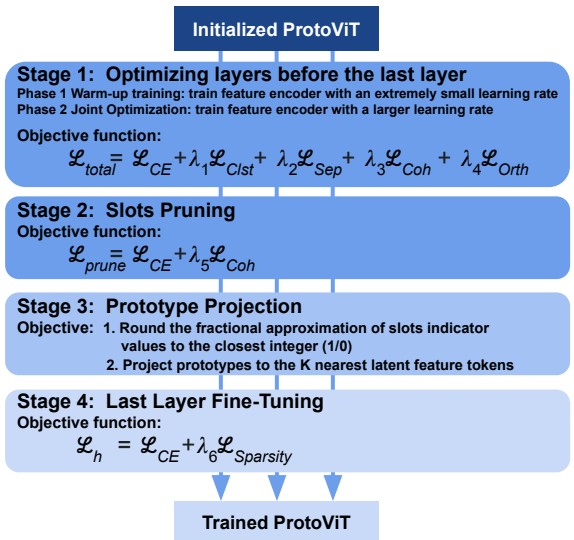

Figure 5: The procedure for training ProtoViT. The objectives for each stage are shown.

Table 4: Parameter Settings for ProtoViT

| Parameter | Weight |
|---|---|
| Cross Entropy Weights | $1.0$ |
| Cluster Loss | $-0.8$ |
| Separation Loss | $0.09$ |
| $L_1$ Norm Loss | $1 \times 10^{-2}$ |
| Orthogonality Loss | $1 \times 10^{-3}$ |
| Coherence Loss | $3 \times 10^{-3}$ |
| Coherence Loss for Slots Pruning | $5 \times 10^{-5}$ |
| Sigmoid Temperature $\tau$ | $100$ |

Table 5: Training Schedule for ProtoViT

| Training Stage | Model Layers | Learning Rate | Duration |
|---|---|---|---|
| Optimization Warm-up Stage | feature encoder $f$ | $1 \times 10^{-7}$ | |
| | prototype layer $p$ | $3 \times 10^{-3}$ | 5 epochs |
| Optimization Joint Stage | feature encoder $f$ | $5 \times 10^{-5}$ | |
| | prototype layer $p$ | $3 \times 10^{-3}$ | 10 epochs |
| Slots Pruning Stage | slots parameters | $8 \times 10^{-5}$ | 10 epochs |
| Fine-tuning Stage | last layer | $1 \times 10^{-4}$ | 15 epochs |

## C More discussion on the greedy matching layer

Intuitively, a non-deformed prototype $\mathbf{p}_j$ consists of $K$ non-overlapping sub-prototypes $\mathbf{p}_j^k$ with rigid adjacency (i.e., a rectangular shape). These non-deformed prototypes are compared with $K$ non-overlapping latent feature tokens $\mathbf{z}_f^i$ in the same geometric shape (i.e., in a rectangle). To deform such a prototype, we could treat each sub-prototype vector $\mathbf{p}_j^k$ as an independent "prototype" to train. Then, the prototype $\mathbf{p}_j$ is naturally deformed into $K$ independent sub-prototypes $\mathbf{p}_j^k$ that move freely in the latent space. However, this naïve approach could lead to significant overlap among the sub-prototypes. Although this could be mitigated by incorporating a unit-wise orthogonality loss during the training phase to encourage dissimilarity among the sub-prototypes, similar to the orthogonality loss outlined in Eq. 5, such a unit-wise orthogonality loss is in conflict with the objectives of the coherence loss defined in Eq. 4, which encourages the sub-prototypes $\mathbf{p}_j^k$ to remain neighboring in cosine distance to preserve the semantic coherence of each prototype $\mathbf{p}_j$. Thus, an algorithm that could provide unit-wise and non-overlapped matching is ideal, and the greedy matching algorithm with adjacency masking meets the needs. An summary of the algorithm for greedy matching with adjacency masking is shown in Alg. 1.

---

**Algorithm 1** Greedy Distance Matching Algorithm

---

**Require:** Input latent feature tokens $\mathbf{z}$, and prototypes $\mathbf{p}$,
 1: Compute cosine similarity between latent feature tokens and k sub-prototypes.
 2: Initialize masks in shape of latent feature tokens, and sub-prototypes to include all possible pairs
 3: **for** k in K slots of sub-prototypes **do**
 4:     Mask out non-adjacent and selected patches by replacing their similarity scores with a high negative value
 5:     Identify the closest latent feature token for each remaining sub-prototype
 6:     Select the pair that has the highest cosine similarity out of all identified pairs
 7:     Update all masks based on the selected pair
 8:     Track the sequence of selected sub-prototypes for reordering
 9: **end for**
10: Sort the pairs and the corresponding cosine similarity based on its original sub-prototype order

---

## D More discussions on adaptive slots mechanism

As discussed in the Sec. 3.3, The adaptive slots mechanism consists of learnable vectors $\mathbf{v} \in \mathbb{R}^{m \times K}$ and a sigmoid function with a hyper-parameter temperature $\tau$. The vectors $\mathbf{v}$ are sent to the sigmoid function to approximate the indicator function as $\tilde{\mathbb{1}}_{\{\text{Include } \mathbf{p}_j^k\}} = \mathbf{Sigmoid}(\mathbf{v}_j^k, \tau)$. Each $\tilde{\mathbb{1}}_{\{\text{Include } \mathbf{p}_j^k\}}$ is an approximation of the indicator for whether the $k$-th sub-prototype will be included in the $j$-th prototype $\mathbf{p}_j$. To be specific, we define the approximated indication function $\tilde{\mathbb{1}}_{\{\text{Include } \mathbf{p}_j^k\}}$ with temperature $\tau$ as:

$$\tilde{\mathbb{1}}_{\{\text{Include } \mathbf{p}_j^k\}} = \frac{1}{1 + e^{-\mathbf{v}_j^k \tau}} \tag{9}$$

By the range of the sigmoid function, we are able to approximate the indicator function with a high temperature $\tau$. Fig. 6 shows an example of different choices of $\tau$. As the temperature value goes up, the sigmoid function more closely approximates the behavior of the indicator function. Thus, we

picked a sufficiently large $\tau$ for approximation, and later rounded the values to the closest integer (i.e 0 or 1) to serve as the actual slot indicator for each sub-prototype.

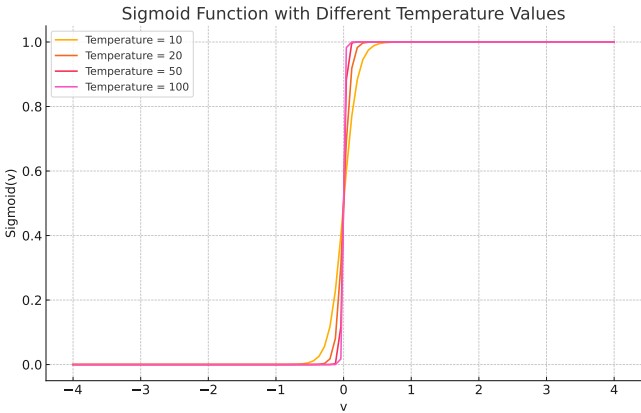

Figure 6: Plot of sigmoid function with different choices of temperature value $\tau$

# E   Ablation studies

This section details ablation studies on the class token, coherence loss, and adjacency mask. We performed the ablation studies with backbone DeiT-Small with $r = 1$ and $K = 4$ for the adjacency mask and number of sub-prototypes respectively.

Table 6: Ablation study on ProtoViT. Model is compared with $K = 4$, and $r = 1$ for slots and adjacency mask settings. We use Deit-Small as the backbone, and the model is trained on CUB-200-2011 dataset. For the current setting of ProtoViT, we included class token, coherence loss, and adjacency mask along with the greedy matching algorithm.

| greedy matching | class token | coherence loss | adjacency mask | Accuracy [%] | PLC [%] |
|---|---|---|---|---|---|
| Yes | No | No | No | $84.32 \pm 0.10$ | $31.02 \pm 0.1$ |
| Yes | Yes | No | No | $85.54 \pm 0.20$ | $35.88 \pm 0.8$ |
| Yes | Yes | Yes | No | $85.13 \pm 0.17$ | $32.98 \pm 0.6$ |
| Yes | Yes | No | Yes | $85.45 \pm 0.30$ | $31.67 \pm 0.2$ |
| Yes | No | Yes | Yes | $84.12 \pm 0.05$ | $17.83 \pm 1.5$ |
| **Yes** | **Yes** | **Yes** | **Yes** | $\mathbf{85.37 \pm 0.13}$ | $\mathbf{21.68 \pm 3.1}$ |

## E.1   Class-token

In this section, we performed an ablation study on the class token. In the ablated model ProtoViT (patch tokens only), the latent feature tokens $\mathbf{z}_f$ are defined as the latent patch tokens $\mathbf{z}_{patch}$. The class token $\mathbf{x}_{class}$ and its corresponding component in position embedding $\mathbf{E}_{pos}$ are excluded in training. As shown in Appendix Table. 6, by incorporating the class token into the latent feature tokens, the performance of the model generally increased by $1\%$ with and without the adjacency mask and coherence loss. Though we observe a drop in performance by excluding the class token and the corresponding component in the position embedding, the ablated model still achieves comparable performance to the other prototype-based ViTs and the black-box backbone.

**Saliency test:**   Subtracting out the class token may have interesting implications for the class specificity of our prototypes. We expect subtracting the class token out operates as a kind of focal similarity, where each token (after taking the difference) represents the unique information available in that position which is relevant to the target class. Prototypes will, then, learn to represent unique positional information that is relevant to the predicted class. As such, we might expect prototypes

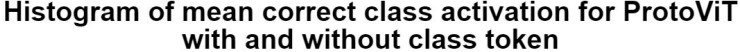

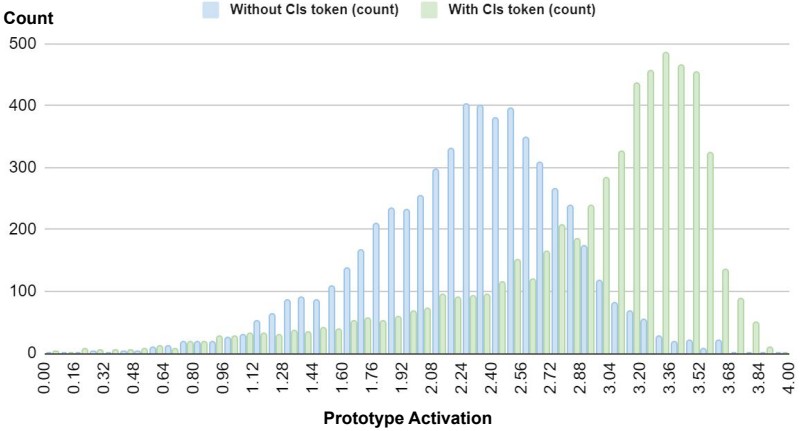

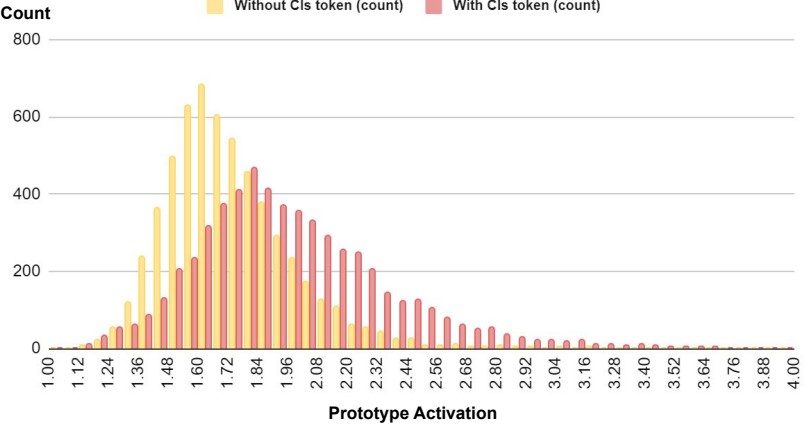

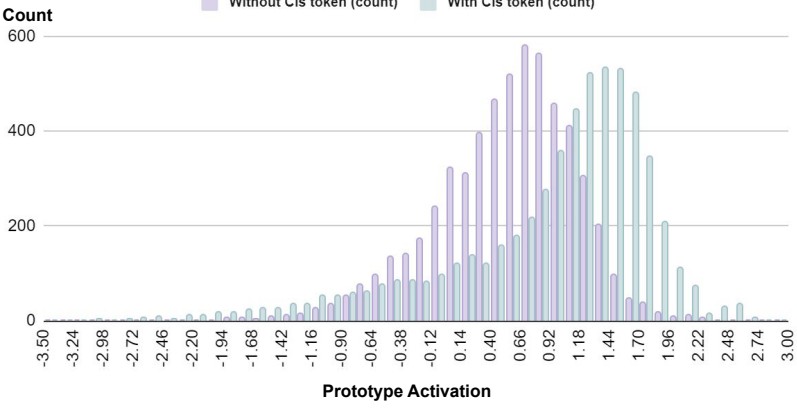

Figure 7: Histogram analysis of mean activation for ProtoViT with and without class token.

to be more tightly tied to their class than they would be if they were learning with simple arbitrary tokens. We thus expect an improvement in class-specific saliency of the prototypes.

To test if class token improved the saliency of prototype to its assigned class, we performed the analysis on correct and incorrect class similarity scores. If prototypes are more specific to their assigned class when including the class token, we would expectthe gap between the mean activation for prototypes of the correct class (summed cosine similarity scores as defined in Eq. 2) and the largest mean activation for prototypes of an incorrect class to be larger. To test this, we randomly pick a model with the class token and one without the class token and evaluate each model on the test set of CUB200-2011[43]. For each test image, we average over the correct class prototypes activations, and denote this average as $\mathbf{g}_{cor}^{abl}$ for the model trained with patch-tokens only, and $\mathbf{g}_{cor}^{cls}$ for the model trained with class tokens. Similarly, we denote the largest mean activations from the incorrect class prototypes as $\mathbf{g}_{incor}^{cls}$, and $\mathbf{g}_{incor}^{abl}$ for the models with and without class token respectively. We use $\delta^{cls}$ and $\delta^{abl}$ to denote the difference between the two measures for the two models. As shown in Appendix Fig. 7, there is a clear shift in distribution between the two models for all three measures. When we include the class token, prototypes tend to activate more highly on the correct class, and the gap between the mean correct class activation and the highest mean incorrect class activation tends to be larger.

We further perform one sided t-tests to test whether $\mathbf{g}_{cor}^{cls} - \mathbf{g}_{cor}^{abl}$ and $\delta^{cls} - \delta^{abl}$ are significantly greater than 0. As shown in Appendix Table. 7, including the class token statistically significantly increases each measure relative to a model trained with patch token only. That is, including the class token improves the saliency of encoded features.

## E.2 Adjacency mask

As introduced in Sec. 3.3, the adjacency mask is designed to ensure that sub-prototypes are geometrically adjacent. Without the adjacency mask, sub-prototypes are able to match with latent feature tokens from anywhere in the image.As indicated in Appendix Table 6, removing the adjacency mask typically results in a slight improvement in performance. This outcome is expected since prototypes are learned for the maximum possible performance with fewer constraints. However, removing the adjacency mask tends to damage the semantics of the model's prototypes. For instance, as demonstrated in Appendix Fig. 8, without the adjacency mask, the model might correctly identify the beak of a least tern in one image patch but erroneously relate other prototypical sub-parts to the bird's feet in another patch. Although the slim feet of the least tern indeed looks similar to the slim beak on the scale of the sub-prototype, such comparisons are not meaningful. In contrast, implementing the adjacency mask in ProtoViT effectively prevents such issues by enforcing geometric adjacency among the sub-prototypes. Furthermore, without the adjacency mask, the model may learn sub-prototypes that, though visually similar to other sub-prototypes, are fundamentally different, introducing noisy representations and disrupting the coherence of prototypical features. For instance, as demonstrated in Appendix Fig. 8, both models misclassified a test image of a black tern as a pigeon guillemot which also has red feet. Without the adjacency mask, the prototypical feature capturing the red feet of the pigeon guillemot also mistakenly includes reflections of the red feet in the water. Although the water reflection looks similar to the actual red feet captured by the other sub-prototypes, this misrepresentation undermines the coherence of the feature representation. On the other hand, ProtoViT compares the feet of the black tern to the red feet of the pigeon guillemot. Thus, the use of an adjacency mask is essential for preserving the coherence of prototypical feature representations.

Table 7: Results of the t-test over correct class activation with and without class token, and the difference between the correct and incorrect class activation with and without class token. The p-values for both tests are $\sim 0$.

| Target | Mean $\pm 1.96$ Std. | T-stats against 0 |
|---|---|---|
| Correct Class Activation ( $\mathbf{g}_{cor}^{cls}$ - $\mathbf{g}_{cor}^{abl}$ ) | $0.625 \pm 0.011$ | 116.3 |
| Correct and Incorrect Class Activation Difference ($\delta^{cls}$ -$\delta^{abl}$ ) | $0.375 \pm 0.008$ | 47.1 |

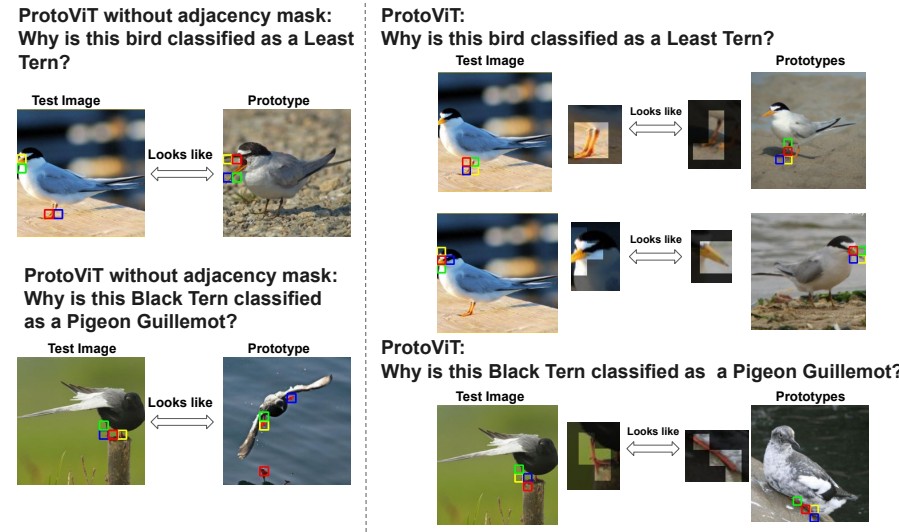

Figure 8: How ProtoViT without adjacency mask makes predictions (left) vs. how ProtoViT makes predictions. The test image of a least tern is correctly classified by the two models, and the test image of a black tern is classified as a pigeon guillemot by both of the models.

### E.3 Coherence loss

As explained in Eq. 4, coherence loss is used to ensure that the sub-prototypes within each prototypical feature are semantically similar to each other. By design, this loss term helps sub-prototypes to collectively represent a coherent feature. By removing the coherence loss, the prototypical features would still contain diverse parts of features in one prototypical feature, a similar problem that the non-deformed prototypes have, even though the sub-prototypes remain geometrically adjacent. For example, as shown in Appendix Fig. 9, a model trained without coherence loss tends to mix the wing and belly of the Myrtle Warbler into one prototypical feature, while it mixes the head and wing for the prototypical features for Painted Bunting. In comparison, training ProtoViT with coherence loss ensures that all the sub-prototypes collectively represent one prototypical feature, in this case containing the striped belly of the Myrtle Warbler, or the wing and the red lower part of the Painted Bunting.

### E.4 Ablated Misalignment

We computed the Prototype Location Change (PLC) for each ablated model using greedy matching algorithms. Following the PLC metric defined in Sacha et al. (2024) [37], we measured the shift in the 90th percentile of prototype activations across test images. As shown in Table 6, the combination of coherence loss and adjacent masks effectively reduces changes in prototype locations after adversarial attacks. This indicates that these mechanisms promote greater stability in the learned prototypes. Interestingly, incorporating the class token into the latent representations increases PLC. This is expected, as the class token contains more global information, which, while improving overall model performance, leads to larger shifts in prototype locations.

## F   Choices of K sub-prototypes

**Why $K=4$ :**   By the design of the vanilla ProtoPNet[10] and the other non-deformed CNN based prototype models[46, 15, 26], the input images are encoded into latent features $z$ by CNN backbones, where $z \in \mathbf{R}^{7 \times 7 \times d}$ and $d$ denotes the latent dimension varying by different choices of backbones. These models learn prototypical features $p$ of shape $1 \times 1 \times d$ from the latent features. On the other hand, ViT backbones such as CaiT and DeiT encode the input images into $14 \times 14 \times d$ latent feature tokens. To be consistent with existing models, we design the model to learn prototypes of shape $2 \times 2 \times d$ from the $\mathbf{R}^{14 \times 14 \times d}$ latent features, so that each prototype corresponds to the same proportion of the input image. Using the greedy matching algorithm, we can move away from

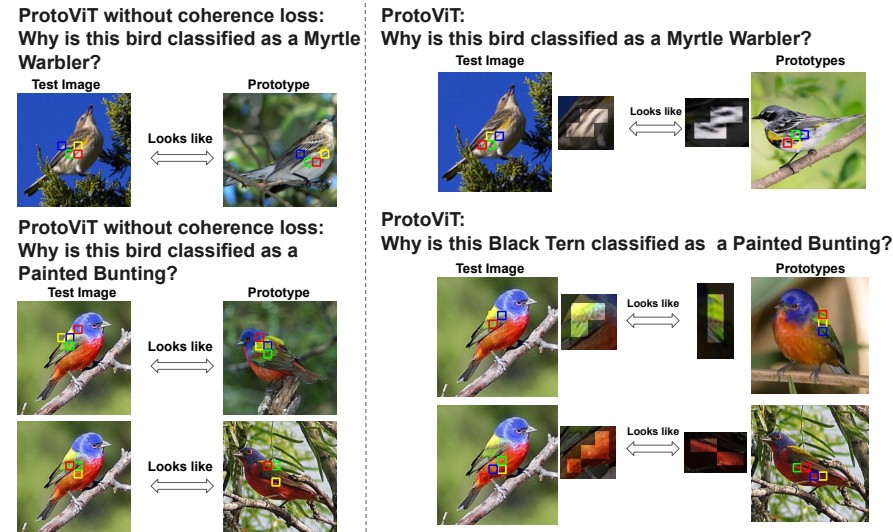

Figure 9: How ProtoViT without coherence loss make predictions (left) vs. how ProtoViT makes predictions. The given test image of Myrtle Warbler and Painted Bunting are correctly classified by both models.

learning fixed $2 \times 2$ rectangular prototypes and instead have **adaptively shaped prototypes** consisting of four sub-prototypes. It is worth noting that prototype-based models with CNN backbones heavily rely on up-sampling for prototype visualizations. This process can introduce errors, leading to the use of the top 5% most similar regions for visualization, which results in irregularly sized bounding boxes, as shown in the top row of Fig. 1. In contrast, with ViT backbones, we are able to visualize the exact image patches that the prototype projected to, and thus produce more precise and accurate prototype visualizations.

**Other choices of $K$ :** Although we selected $K = 4$ to maintain consistency with other existing prototype-based models, alternative values of K are also viable. The model performance for $K = 5$ and $K = 6$ settings can be found in Appendix Table. 8. We observed that the model performs best with $r = 2$ when $K = 5$ is used, and with $r = 3$ when $K = 6$. As shown in the table, the models with different choices of $K$ perform similarly to the setting with $K = 4$. Appendix Fig. 10 and Fig. 11 show examples of reasoning process for ProtoViT($K = 5, r = 2$) and ProtoViT($K = 6, r = 3$) respectively. As shown in the figures, though having more sub-prototypes is helpful to capture larger featuires such as the tail of the Lincoln Sparrow and Forster Tern in Fig. 10 and Fig. 11 respectively, because of variations in scale, many features such as the beak and the head in 11 do not always need that many sub-prototypes. Local and global analysis for ProtoViT($K = 5, r = 2$) and ProtoViT($K = 6, r = 3$) are shown in Fig. 13 and Fig. 13 respectively. As shown in the figures, having more sub-prototypes is advantageous in representing features that are large in scale such as the white belly of Sayornis shown in the second row of global analysis in Fig. 12, and brown pattern of Eastern Towhee in the bottom row of Fig. 13. Both the local and global analysis again show that **the prototypes of ProtoViT have strong, consistent semantic meanings.**

It is worth noting that prior prototype-based models could not easily support prototypes with 5 or 6 sub-prototypes. Since methods like Deformable ProtoPNet treat prototypes as convolutional features, the only way to handle prototypes with 5 sub-prototypes would be to form each prototype as a 1 by 5 dimensional convolutional filter. This would allow the model to have 5 sub-prototypes, but would enforce a very strange shape on prototypes (a horizontal line).

# G   Qualitative examples of robustness against perturbations

We provide several instances of the perturbation examples, in which we mask out the region selected by each prototype, as shown in Fig.14. In each row, we mask out all matched locations for a prototype (middle-left column) using a white mask on the black wings and a black mask on the red parts,

Table 8: ProtoViT performance with $K = 5$ and $K = 6$ using DeiT-Small backbone

| # of sub-prototypes | Adjacency mask range $R$ | Accuracy [%] |
|---|---|---|
| 4 | 1 | $85.37 \pm 0.13$ |
| 5 | 2 | $85.33 \pm 0.20$ |
| 6 | 3 | $85.26 \pm 0.15$ |

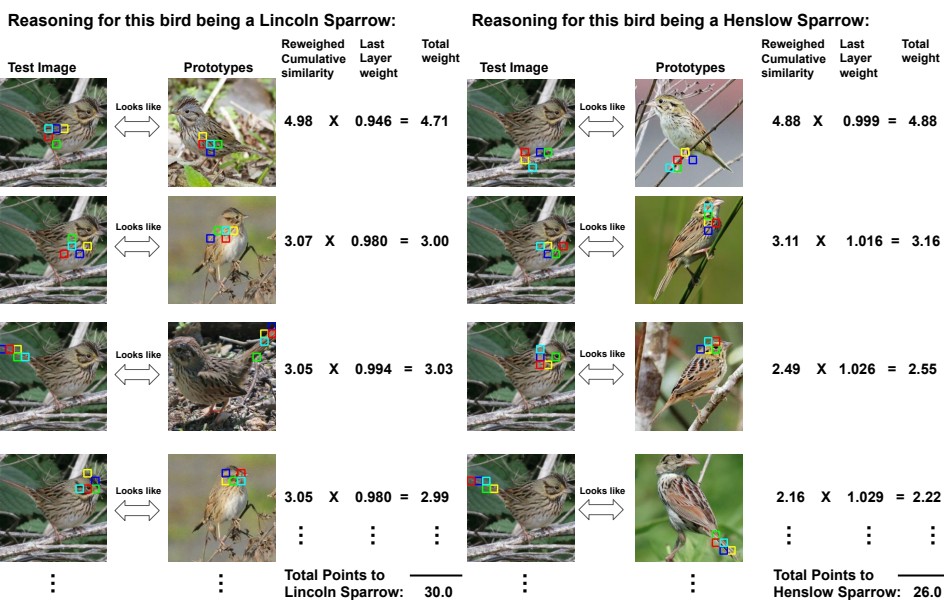

Figure 10: How ProtoViT with $K = 5$ and $r = 2$ using DeiT-Small backbone classifies a test image of a Lincoln Sparrow to the correct class (left), and the second most likely class Henslow Sparrow (right).

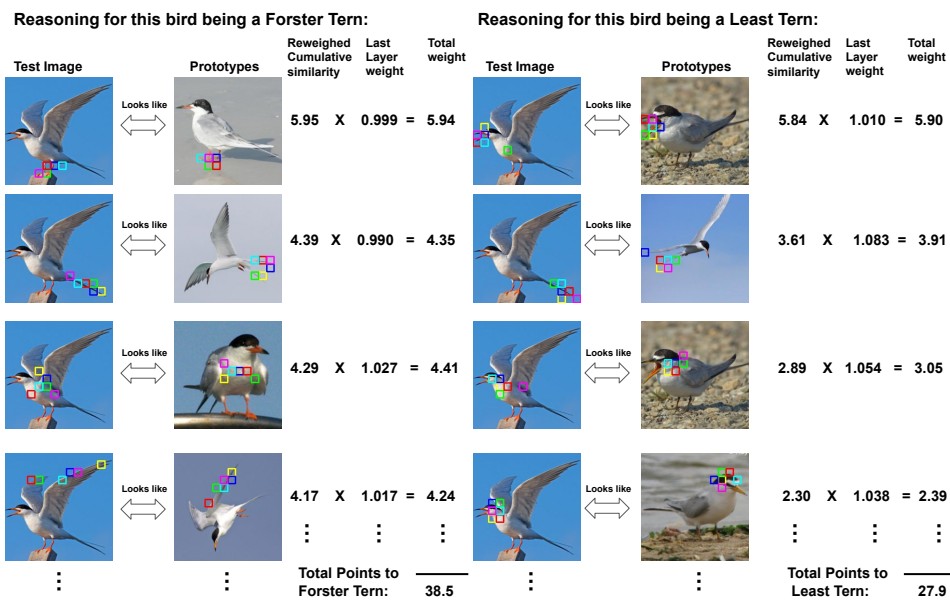

Figure 11: How ProtoViT with $K = 6$ and $r = 3$ using DeiT-Small backbone classifies a test image of a Forster Tern to the correct class (left), and the second most likely class Least Tern (right).

and check where that prototype activates after masking (shown in the leftmost column). We then confirm that the activated region for other prototypes remains reasonable when the mask from another

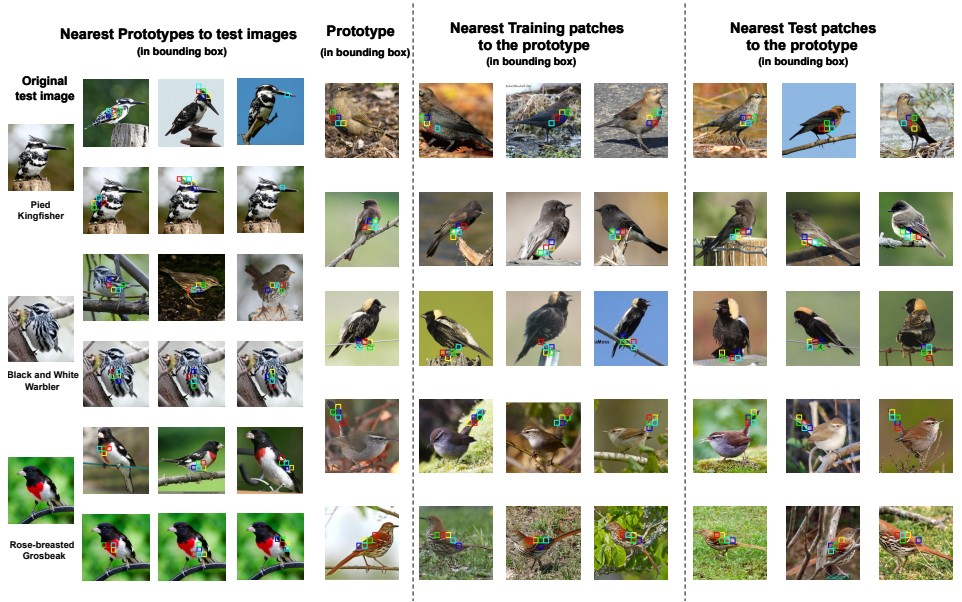

Figure 12: Examples of local analysis (left) and global analysis (right) of ProtoViT with DeiT-Small backbone with $K = 5$ and $r = 2$.

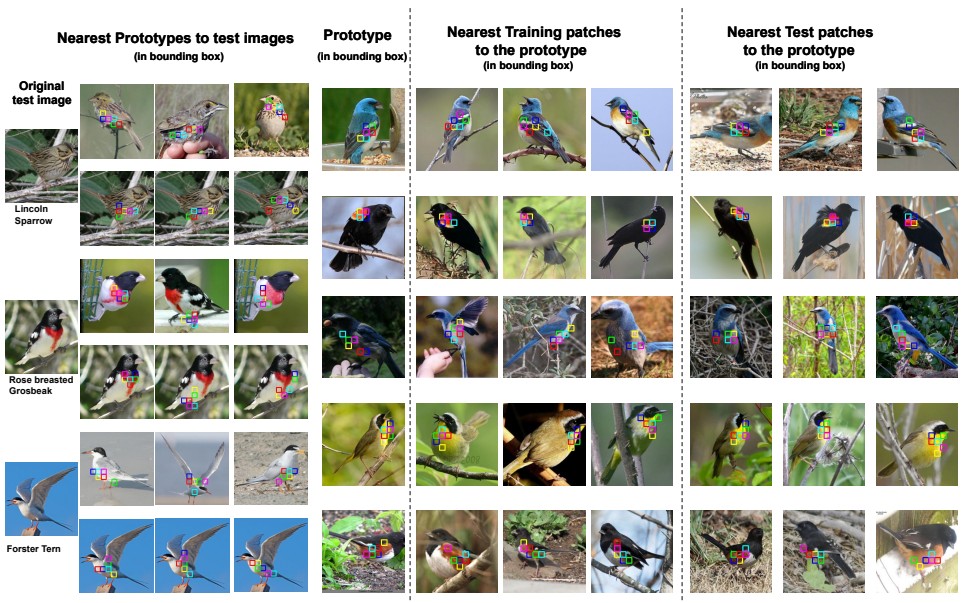

Figure 13: Examples of local analysis (left) and global analysis (right) of ProtoViT with DeiT-Small backbone with $K = 6$ and $r = 3$.

prototype is applied (right two columns). We observed that after removing the preferred region by each prototype, it activates on another reasonable alternative (e.g., a red belly prototype might activate on a red back as a second choice).

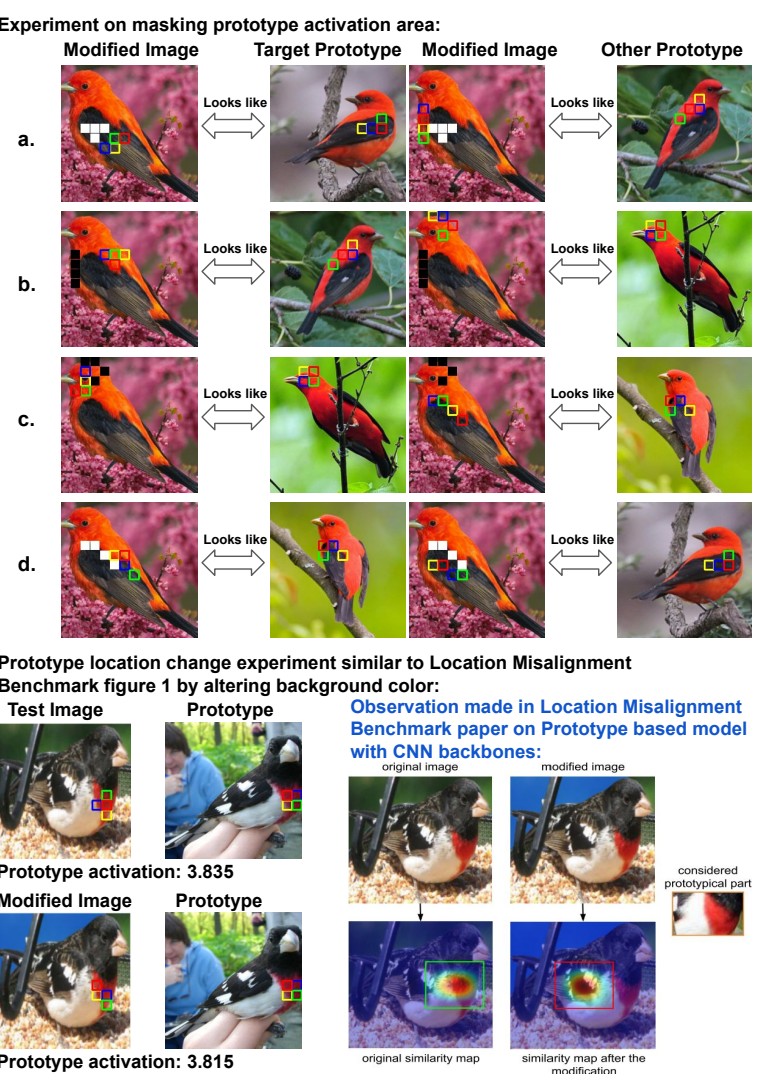

Figure 14: Perturbations (top) and background change (bottom). As shown in the top, our prototypes identify reasonable alternative regions on which to activate after masking the originally most activated regions. As shown in the bottom, altering the background does not substantially impact the activation location of our ViT-based prototype. This shows that our ProtoViT has a better location alignment than the CNN-based prototype models.

# H   More examples of reasoning process

This section provides more examples for the model reasoning process. Fig. 15, Fig. 16, and Fig. 17 demonstrate more examples of the reasoning process of ProtoViT with DeiT-Small backbone. Fig. 18, Fig. 19 and 20 are the examples of reasoning process of ProtoViT with CaiT-backones. Fig. 21, Fig. 22 and Fig. 23 are the examples of reasoning process of ProtoViT with Deit-Tiny backbone. In each case, we again see intuitive reasoning from ProtoViT.

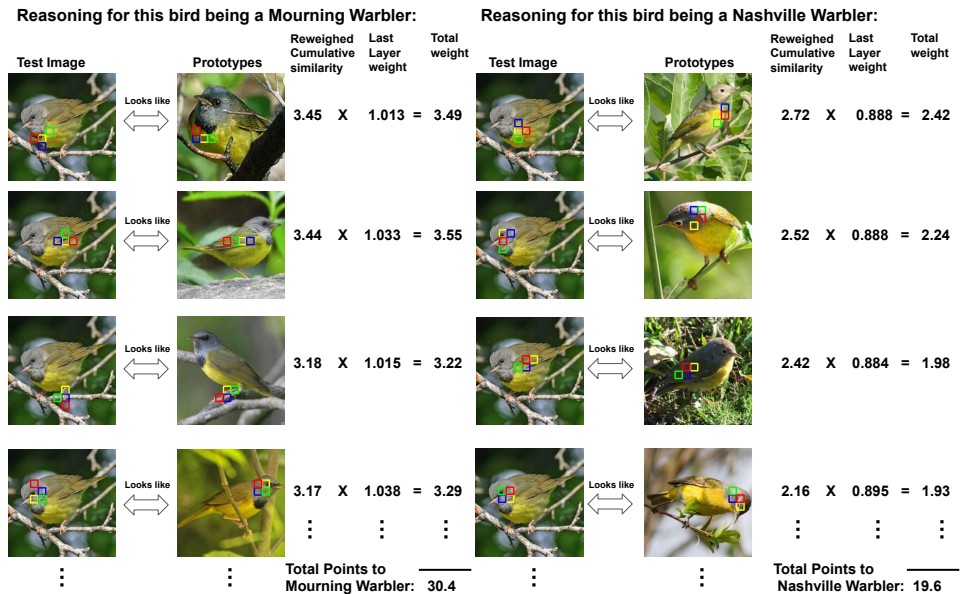

Figure 15: How ProtoViT with DeiT-Small backbone classifies a test image of a Mourning Warbler to the correct class (left), and the second most likely class Nashville Warbler (right).

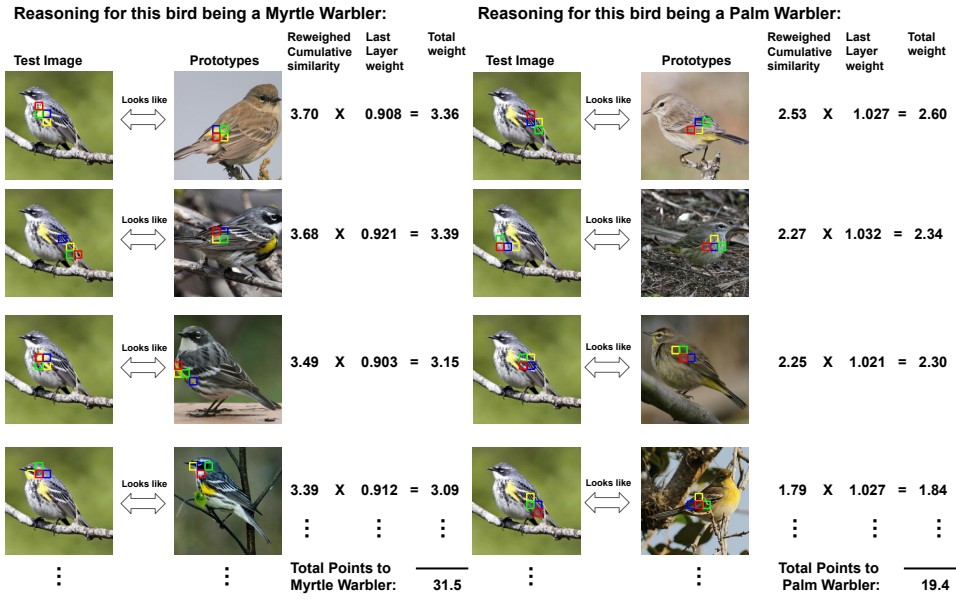

Figure 16: How ProtoViT with DeiT-Small backbone classifies a test image of a Myrtle Warbler to the correct class( left), and the second most likely class Palm Warbler (right).

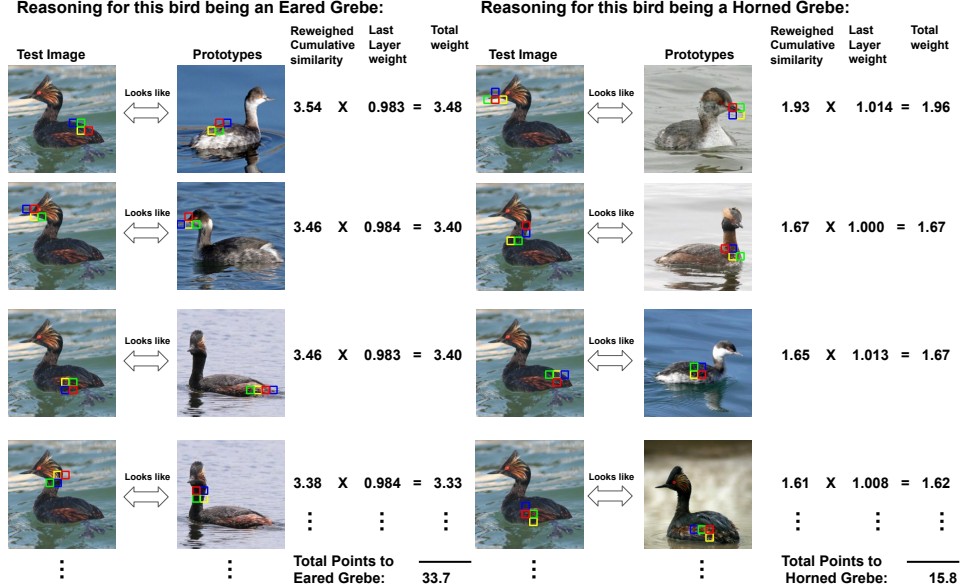

Figure 17: How ProtoViT with DeiT-Small backbone classifies a test image of an Eared Grebe to the correct class (left), and the second most likely class Horned Grebe (right).

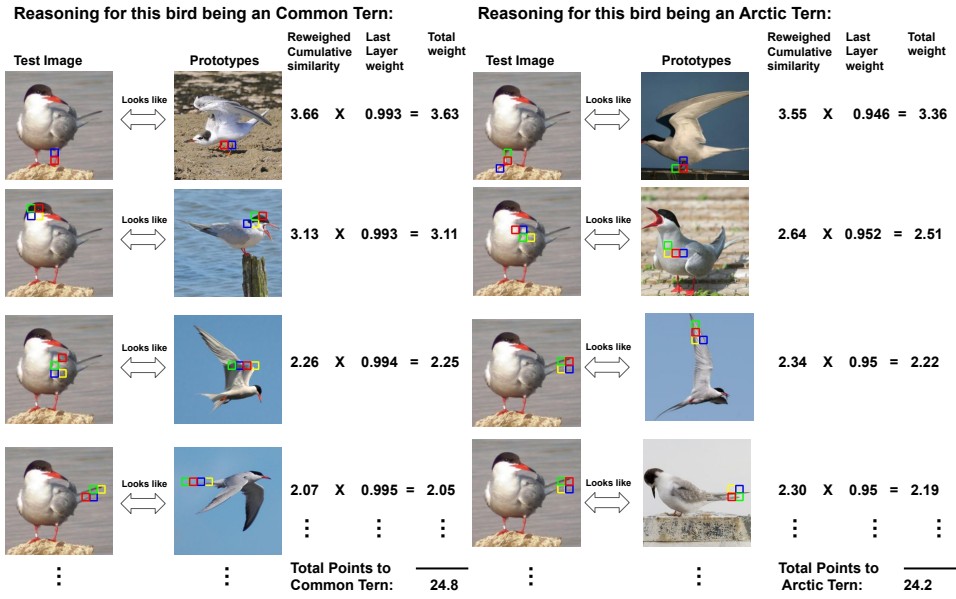

Figure 18: How ProtoViT with CaiT-xxs backbone classifies a test image of a Common Tern to the correct class (left), and the second most likely class Arctic Tern (right).

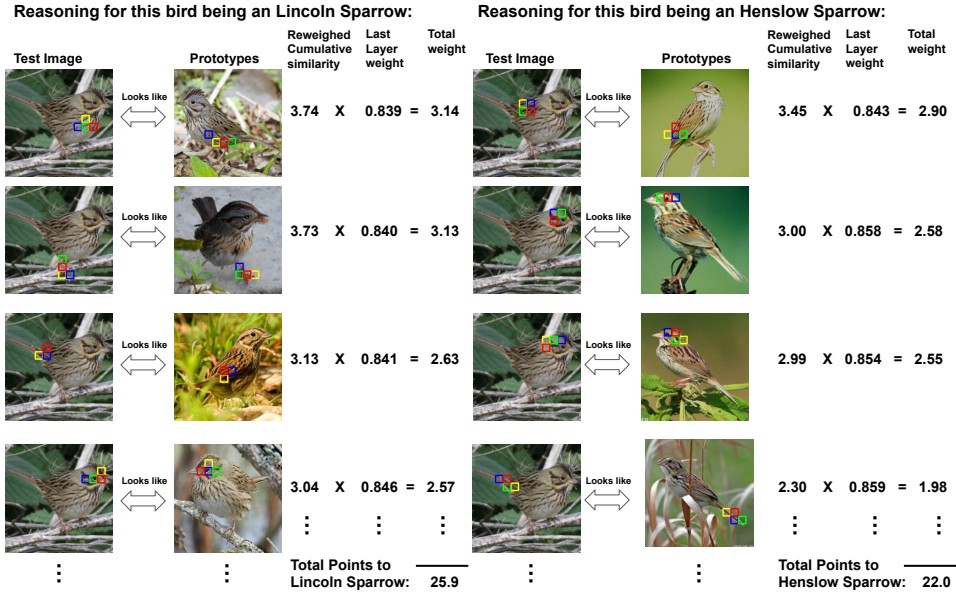

Figure 19: How ProtoViT with CaiT-xxs backbone classifies a test image of a Lincoln Sparrow to the correct class (left), and the second most likely class Henslow Sparrow (right).

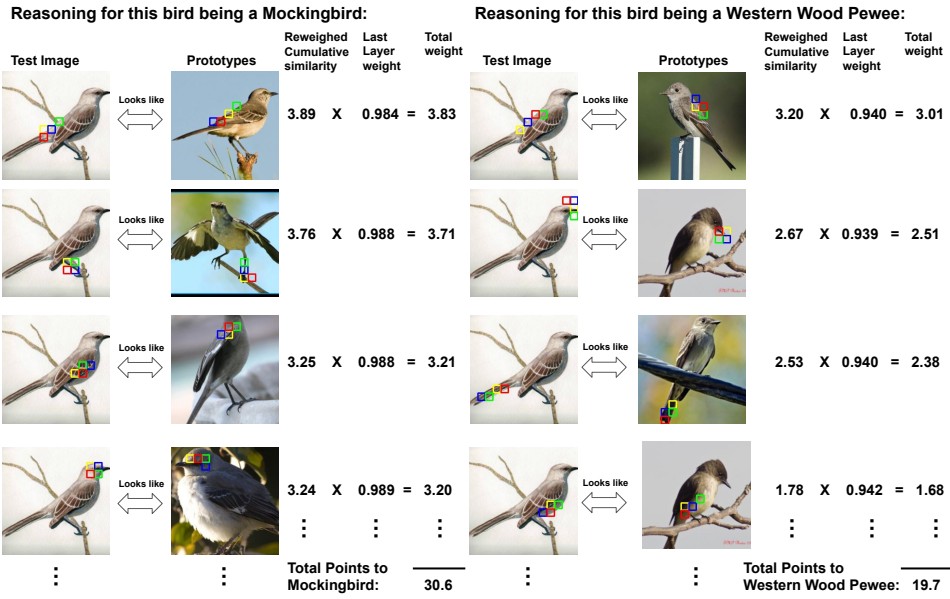

Figure 20: How ProtoViT with CaiT-xxs backbone classifies a test image of a Mockingbird to the correct class (left), and the second most likely class Western Wood Pewee (right).

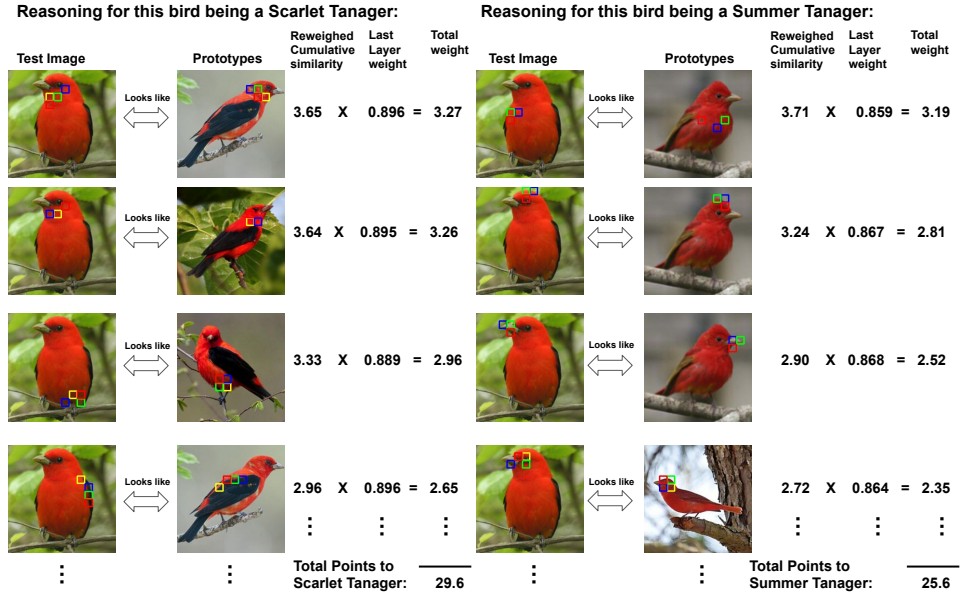

Figure 21: How ProtoViT with Deit-Tiny backbone classifies a test image of a Scarlet Tanager to the correct class (left), and the second most likely class Summer Tanager (right).

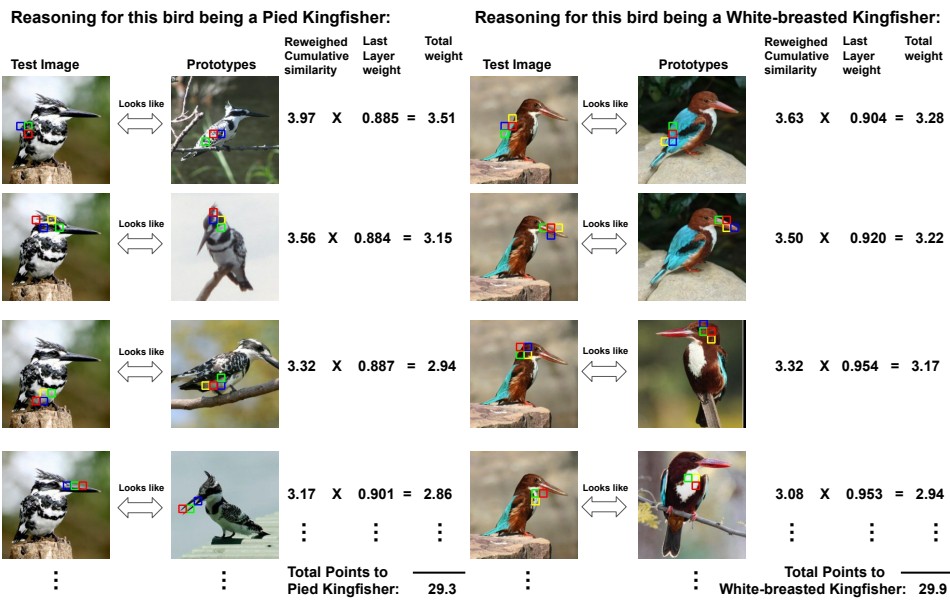

Figure 22: How ProtoViT with Deit-Tiny backbone classifies a test image of a Pied Kingfisher to the correct class (left), and classifies a test image of a White-breasted Kingfisher to the correct class (right).

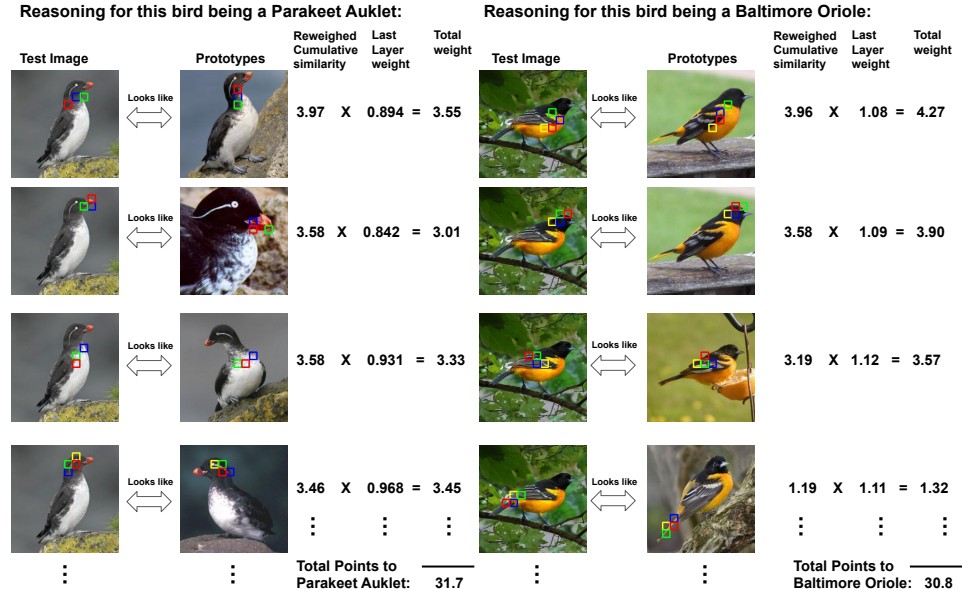

Figure 23: How ProtoViT with Deit-Tiny backbone classifies a test image of a Parakeet Auklet to the correct class (left), and classifies a test image of a Baltimore Oriole to the correct class (right).

# I  More examples of reasoning process for misclassification

In this section, we provide the reasoning process of how our model misclassified a test image of a summer tanager and a slaty backed gull in Fig. 24. We found that the misclassification of the given summer tanager example may be because of data mislabeling in the original dataset. A summer tanager does not have a black colored wing. That test image should indeed belong to the scarlet tanager class as the model predicted. Similar mislabeling cases also happen for Red Headed Woodpecker with image ID Red_Headed_Woordpecker_0018_183455 and Red_Headed_Woordpecker_0006_183383, as we found out when randomly selecting examples to present for the paper. On the other hands, misclassifications can be contributed by the similarity between different classes. As shown in the bottom of Fig. 24, the West Gull looks very similar to Slaty Gull. And the model's reasoning process indeed shows that the model believes that these two classes are the top two most likely classes for prediction.

These examples showcase the ability of our method to help us understand the reasoning process of the model, not just when it is right, but also when it is wrong.

# J  More examples of analysis

This section provides more examples for the local and global analysis. Fig. 25, Fig. 26, and Fig. 27 are the examples for analysis for ProtoViT with Deit-Small, CaiT-xxs24 and DeiT-tiney backbones respectively. The visualizations demonstrate that **the prototypes of Protovit exhibit consistent, strong semantic meanings across different ViT backbones.**

# K  Details on User Studies

**Overview:**  This section provides details on the user study we conducted. Through our user study, we show that ProtoVit improves both the clarity of reasoning process and coherence of prototypical features relative to ProtoPNet[10] and Deformable ProtoPNet[13]. We randomly selected 10 bird images from test set, and show the comparison with the three most similar prototypes from its top-1 predicted class of the three models. We then presented the test to 10 participants, all of whom attend college in the U.S. and have some background in Machine Learning. We instructed participants to rate how well they understood the models' reasoning process through the presented examples,

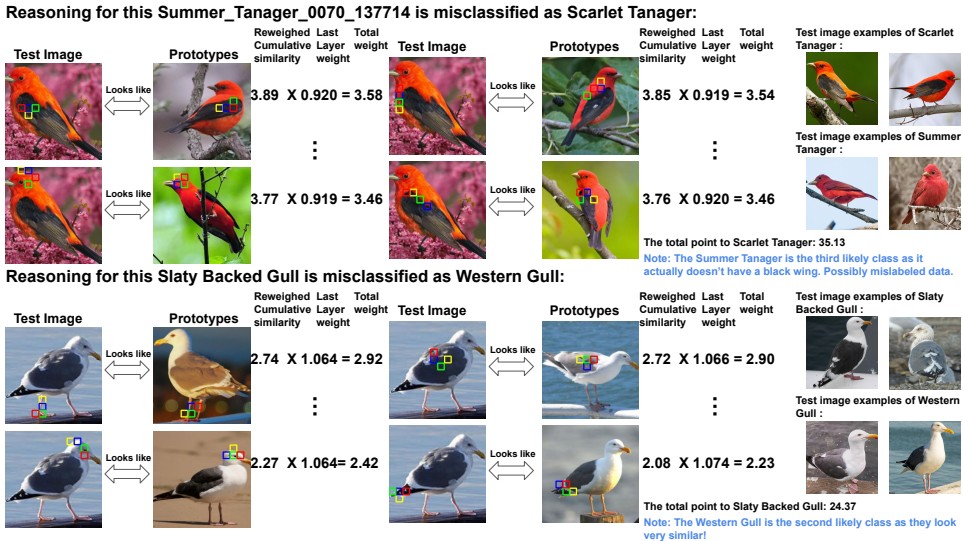

Figure 24: Misclassification examples. Reasoning process of how our model misclassified a test image of a summer tanager (top) and a slaty backed gull (bottom). The top misclassification is due to mislabeling.

and asked them to rate their confidence in their evaluation. Intuitively, the user should easily be able to distinguish which feature the model is comparing to, if the model has superb clarity. At the end of the experiment, we asked participants to rank the three models based on overall clarity of reasoning as well as coherence of prototypical features learned. Coherence of feature refers to when the visualization represents one, and only one feature. Some sample questions are shown in Appendix. Fig. 28. Although this task does not pose any risk to the participants, they were nevertheless informed of their rights, and were asked for consent before data collection. The participants took on average 10 to 20 minutes to complete their assigned tasks, and were compensated at $30 per hour.

**Result:** To quantify the user study result on model clarity, we assign a score from 4 to 1 as the participant rated their understanding of the model's reasoning from best to bad. Similarly, we assign a score from 4 to 1 for the confidence rating from completely confident to not confident. We then multiply the confidence score and the understanding score to have the total score on the clarity of the model reasoning. The maximum total score for a question is 16, which indicates that the participant has a very clear and confident understanding of the model's reasoning. The minimum total score for a question is 1, which indicates that the participant does not understand the model's reasoning at all. As shown in Table. 9, on average, the participants believe that they understand the reasoning process of ProtoViT the best with the highest confidence. The result of the one-sided t-test on understanding score and total score further illustrate that there is a statistically significant improvement in model clarity of ProtoViT to the vanilla ProtoPNet. Fig. 29 shows the result of participants ranking on the three models based on coherence of prototypical features and the clarity of the reasoning process. Again, ProtoViT is mostly ranked as the best in providing coherent prototypical features and clear reasoning process. It is worth noting that the majority of the participants commented that the prototypical features by ProtoPNet are usually too broad to understand what specific part the model is looking at. Moreover, the prototypical features by Deformable ProtoPNet are less accurate in pinpointing the specific parts. On the other hand, the prototypical features by Protovit can provide more accurate and specific visualizations. It is easy to tell if the prototypical features are representing the head of birds or the feet of the birds.

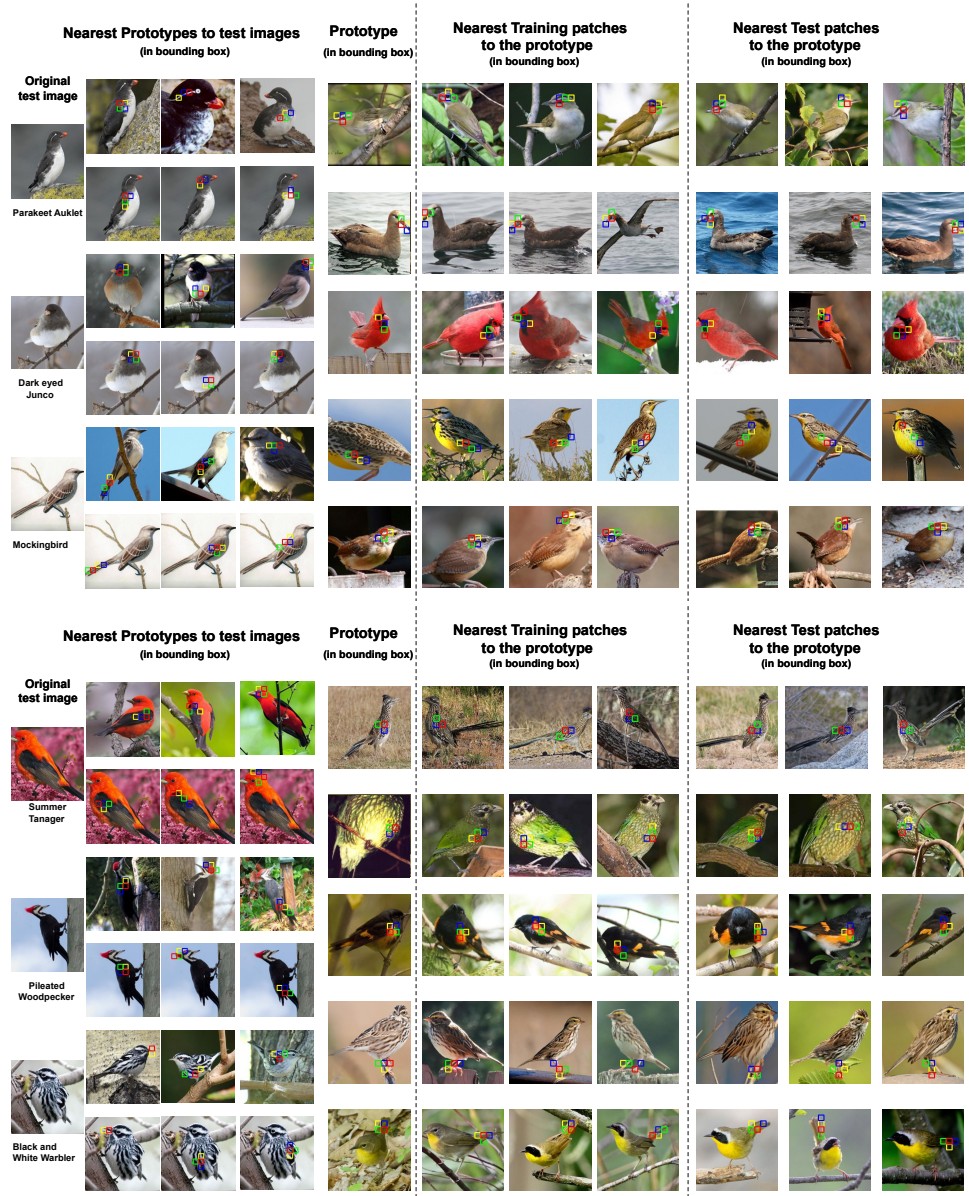

Figure 25: More examples of local analysis(left) and global analysis (right) of ProtoViT with DeiT-small backbone.

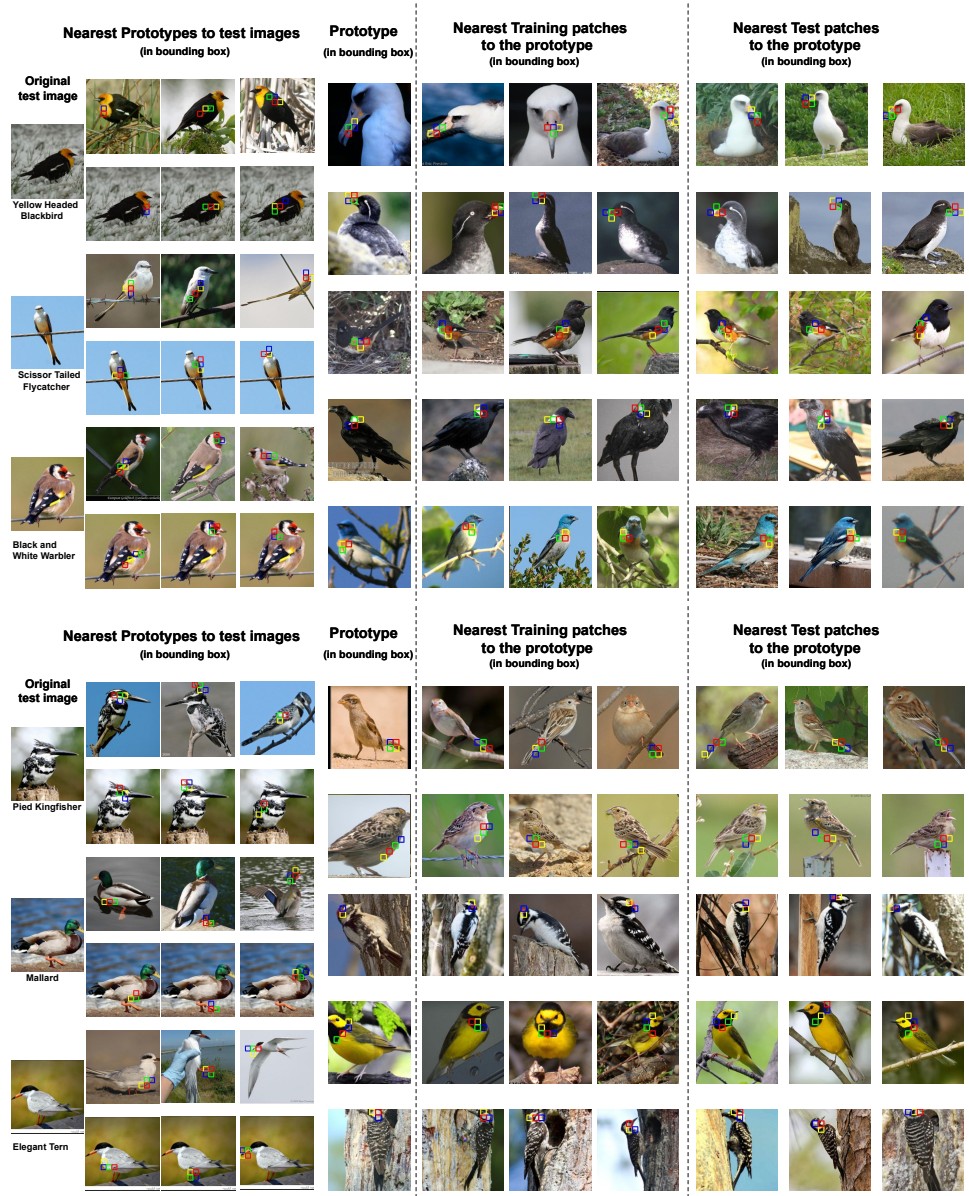

Figure 26: More examples of local analysis (left) and global analysis (right) of ProtoViT with CaiT-xxs24 backbone.

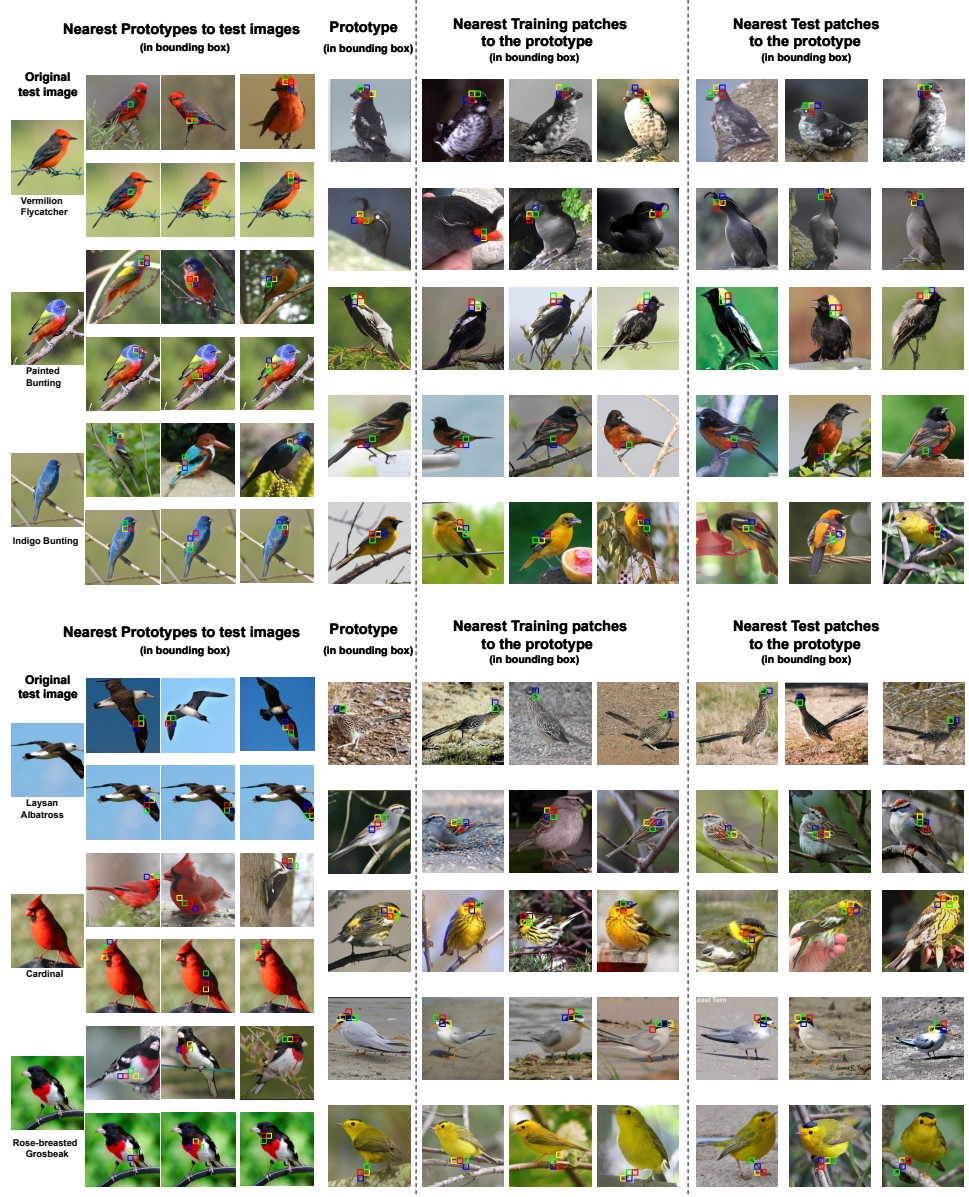

Figure 27: More examples of local analysis (left) and global analysis (right) of ProtoViT with DeiT-Tiny backbone.

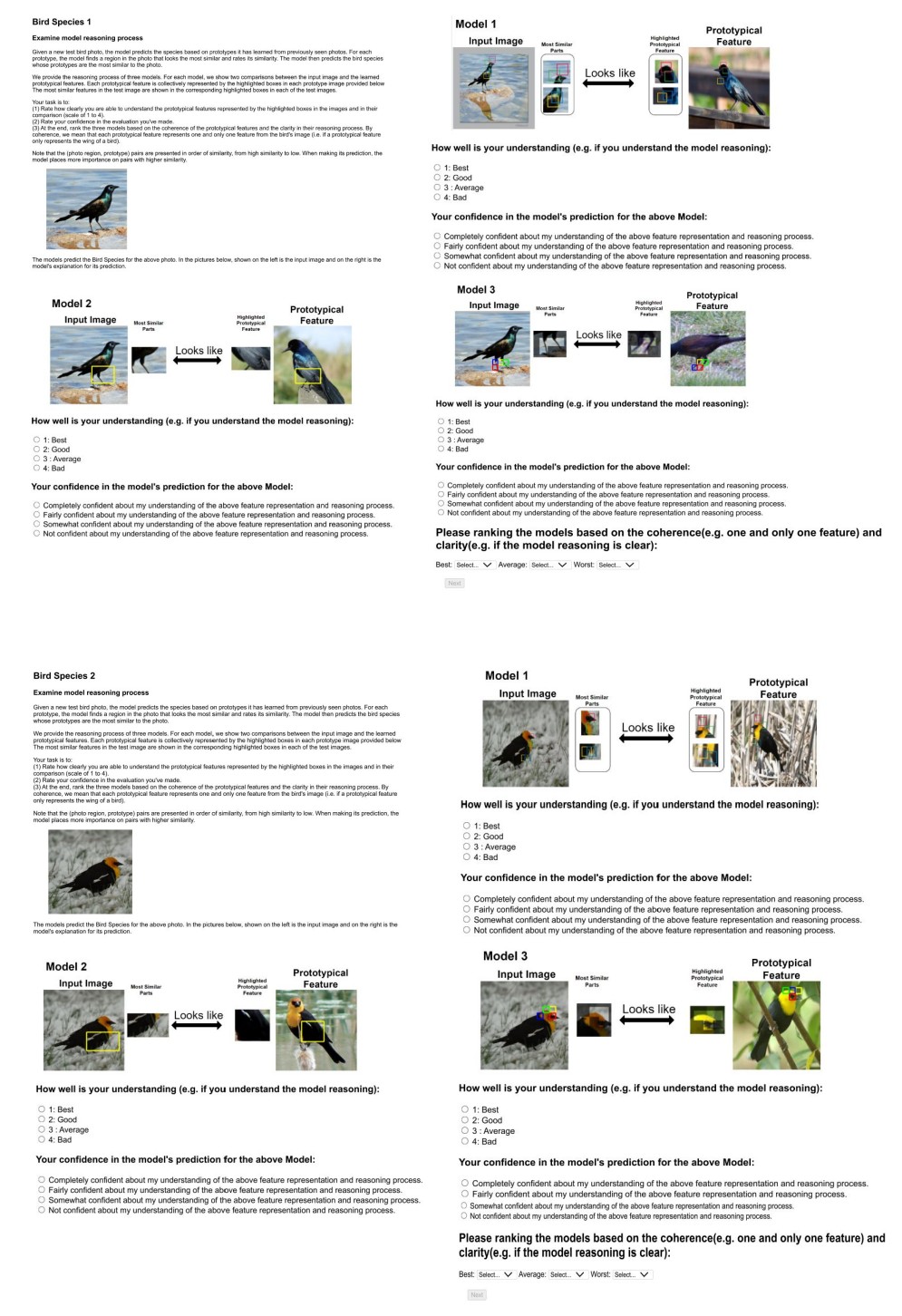

Figure 28: Example of the survey questions. The examples are randomly selected, and the prototype with the highest similarity score for each example is selected.

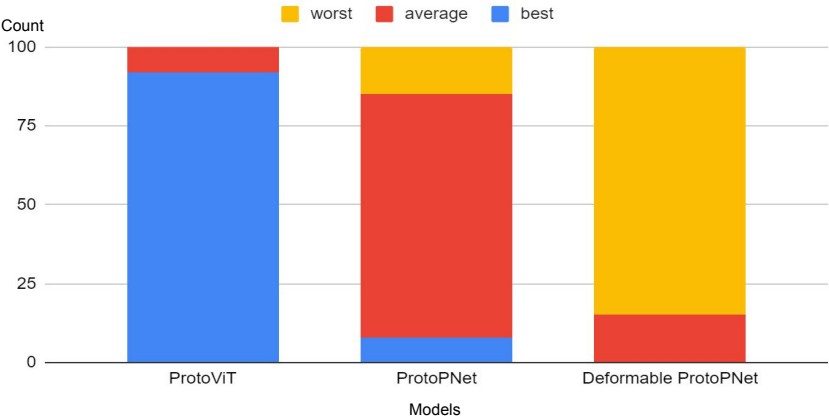

Figure 29: User Study result on model rankings based on its clarity and coherence

Table 9: User Study results on model clarity. We perform one-sided t-test on the total score and understanding score of ProtoViT and ProtoPNet.The result shows that ProtoViT has a statistically significant improvement in clarity of the reasoning process to ProtoPNet.

| Model | Mean understanding score | Mean confidence score | Mean total score |
|---|---|---|---|
| Deformable ProtoPNet [13] | 1.86 | 3.15 | 5.76 |
| ProtoPNet [10] | 2.68 | 3.17 | 8.67 |
| **ProtoViT** | 3.85 | 3.47 | 13.42 |

| T-Test | Mean $\pm$ 1.96 std | T-stats | P-value |
|---|---|---|---|
| ProtoViT total score - ProtoPNet total score | $4.75 \pm 0.894$ | 10.37 | $2.16 \times 10^{-20}$ |
| ProtoViT Understanding score - ProtoPNet Understanding score | $1.17 \pm 0.209$ | 10.89 | $5.75 \times 10^{-22}$ |

## L    Details on Car dataset

This section provides details on the implementation of ProtoViT on a second dataset. The Standford Car dataset [24] contains 8144/8041 images for training and testing from 196 different car models. We performed a similar offline data-augmentation as described in Sec. 4.1. After augmentation, the training set has roughly 1,600 images per class. We assigned the algorithm to choose 10 class-specific prototypes for each of the 196 classes. Each of the prototypes is composed of 4 sub-prototypes. We kept the training schedule and hyper-parameters same as on the bird dataset. The specific training schedule and hyper-parameters settings are documented in Appendix Tables 5 and 4 respectively. Fig. 30 and Fig. 31 show examples of the reasoning process for the DeiT-Small backbone. Fig. 32 and Fig. 33 show examples of the reasoning process for the CaiT-xxs-24 backbone. Fig. 34 and Fig. 35 show examples of the reasoning process for the DeiT-Tiny backbone. Examples of global analysis and local analysis for ProtoViT with different backbones are shown in Fig. 37, Fig. 36, and Fig. 38 respectively.

## M    Broader Impact

Interpretability is an essential ingredient to trustworthy AI systems. Our work successfully integrates one of the most popular and powerful model families into prototype-based networks, which are one of the leading techniques for interpretable neural networks in computer vision. Our technique can

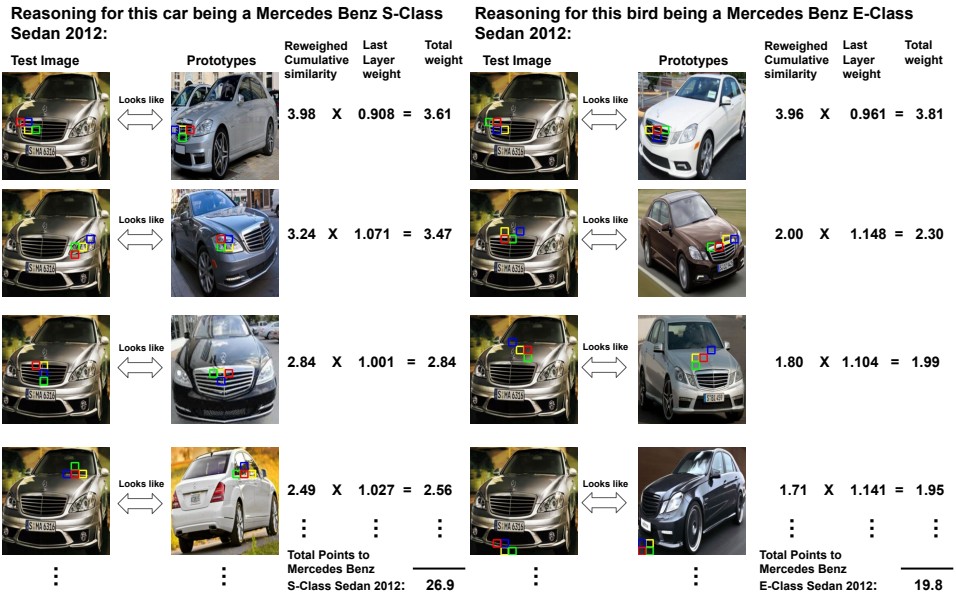

Figure 30: How ProtoViT with Deit-Small backbone classifies a test image of a Mercedes Benz S-class Sedan 2012 to the correct class (left), and the second most likely class Mercedes Benz E-class sedan 2012 (right).

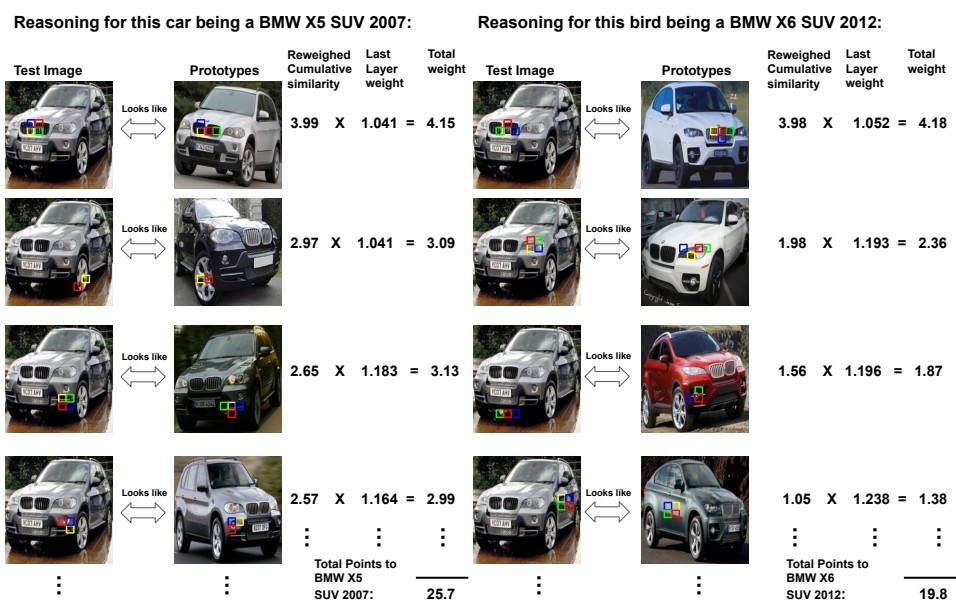

Figure 31: How ProtoViT with Deit-Small backbone classifies a test image of a BMW X5 SUV 2007 to the correct class (left), and the second most likely class BMW X6 SUV 2012 (right).

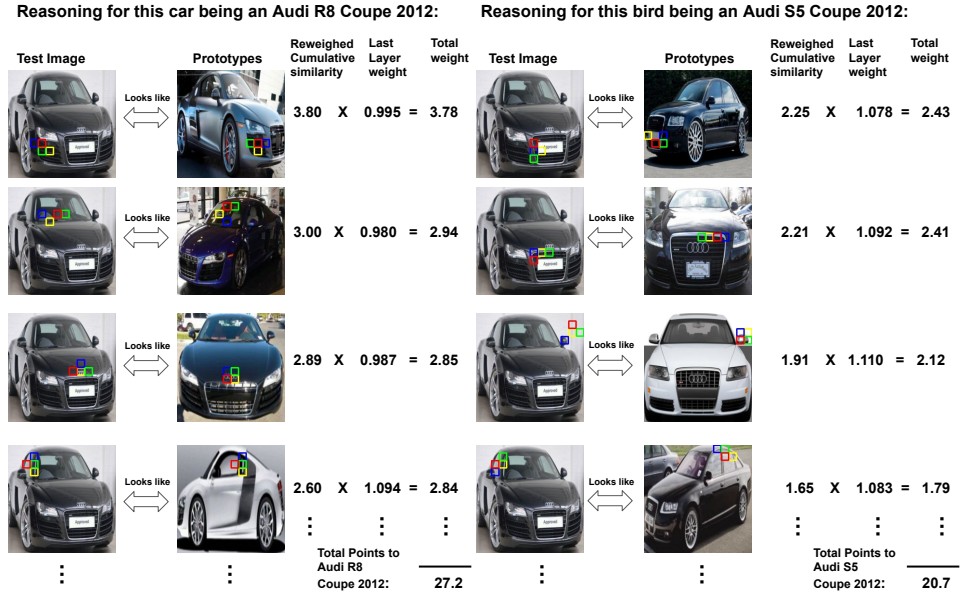

Figure 32: How ProtoViT with CaiT-xxs24 backbone classifies a test image of an Audi R8 Coupe 2012 to the correct class (left), and the second most likely class Audi S5 Coupe 2012 (right).

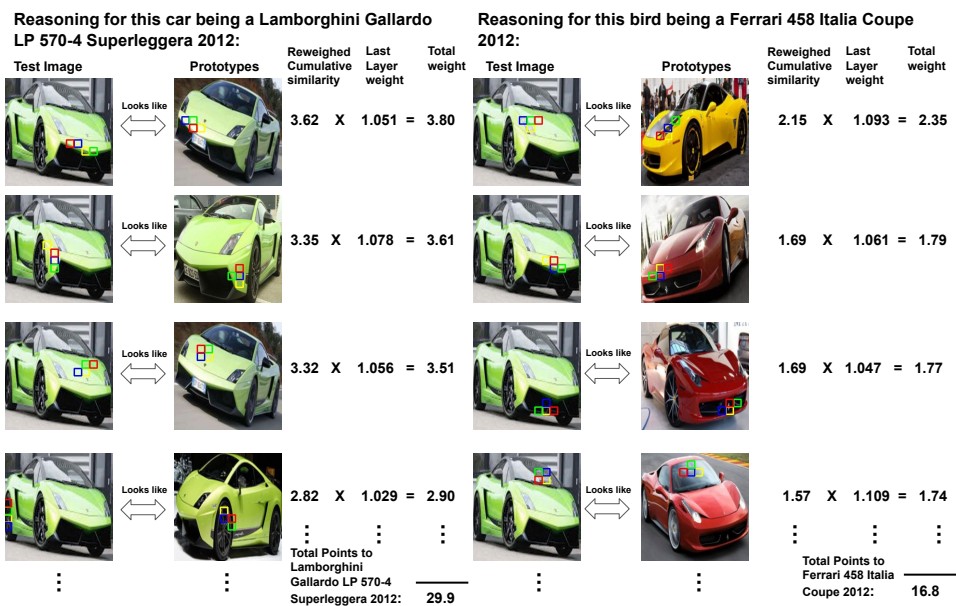

Figure 33: How ProtoViT with CaiT-xxs24 backbone classifies a test image of a Lamborghini Gallardo LP 570-4 Superleggera 2012 to the correct class (left), and the second most likely class Ferrari 458 Italia Coupe 2012 (right).

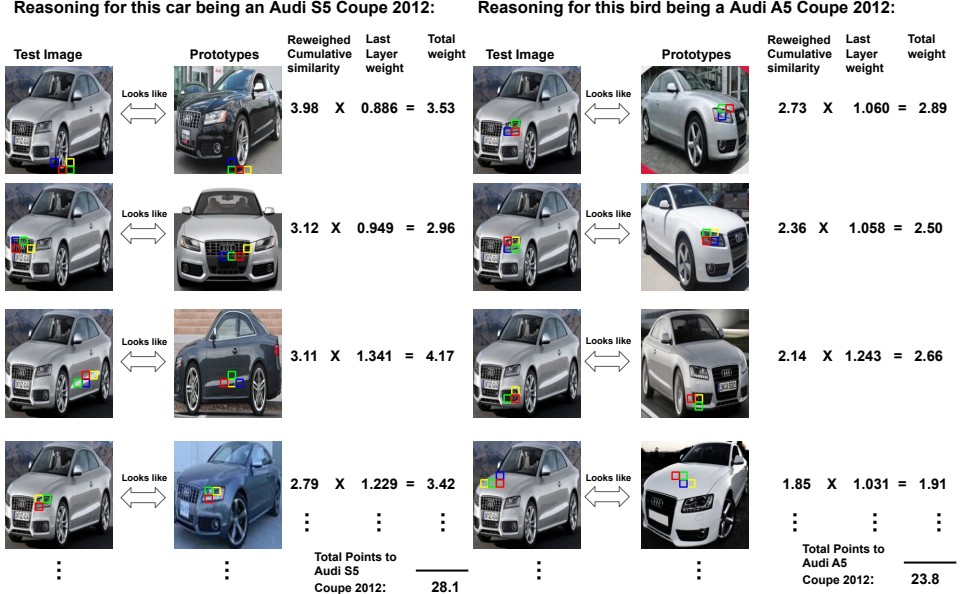

Figure 34: How ProtoViT with Deit-Tiny backbone classifies a test image of an Audi S5 Coupe 2012 to the correct class (left), and the second most likely class Audi A5 Coupe 2012 (right).

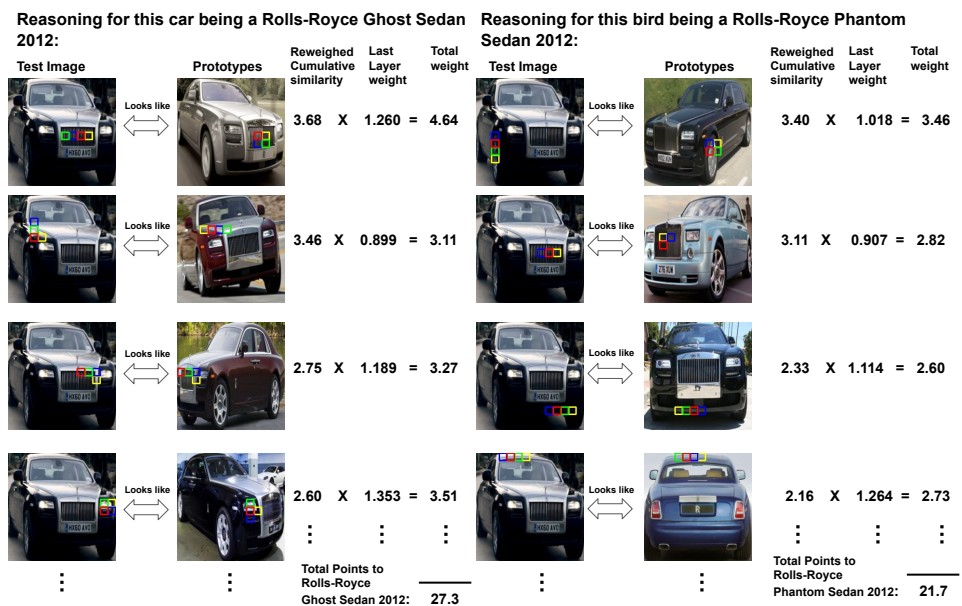

Figure 35: How ProtoViT with Deit-Tiny backbone classifies a test image of a Rolls-Royce Ghost Sedan 2012 to the correct class (left), and the second most likely class Rolls-Royce Phantom Sedan 2012 (right).

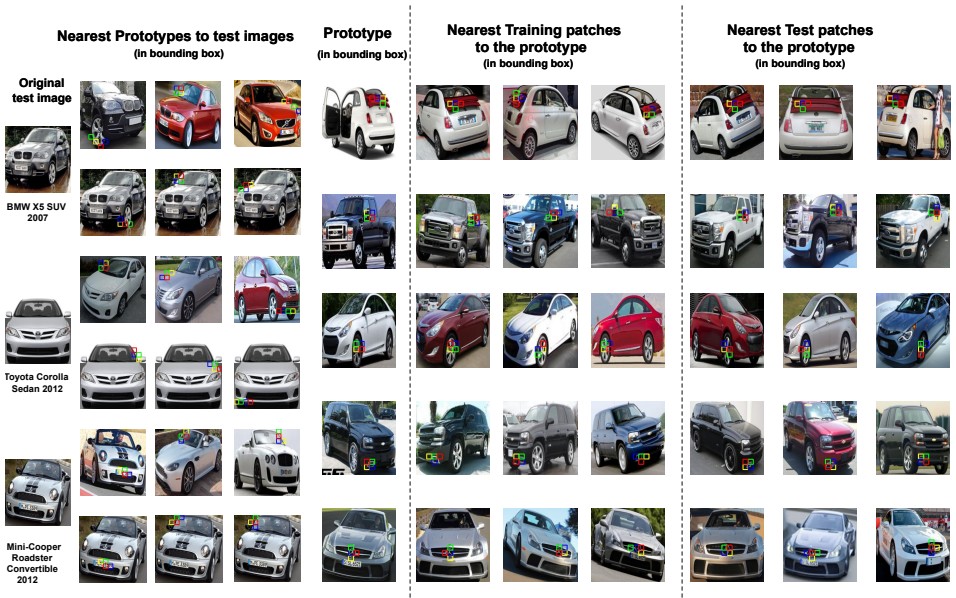

Figure 36: Examples of local analysis (left) and global analysis (right) of ProtoViT with CaiT-xxs24 backbone on the Stanford Cars dataset.

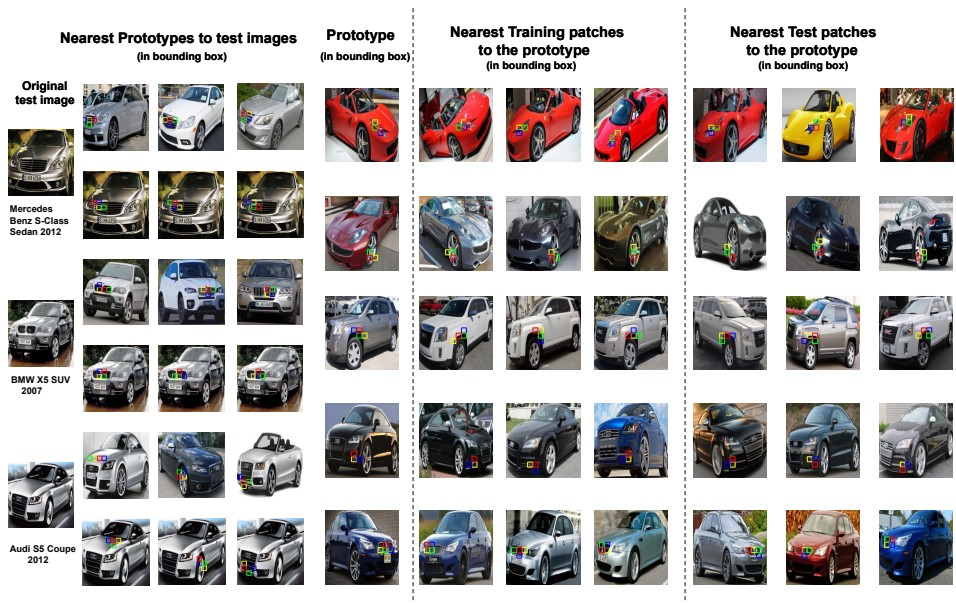

Figure 37: Examples of local analysis (left) and global analysis (right) of ProtoViT with Deit-Small backbone on the Stanford Cars dataset.

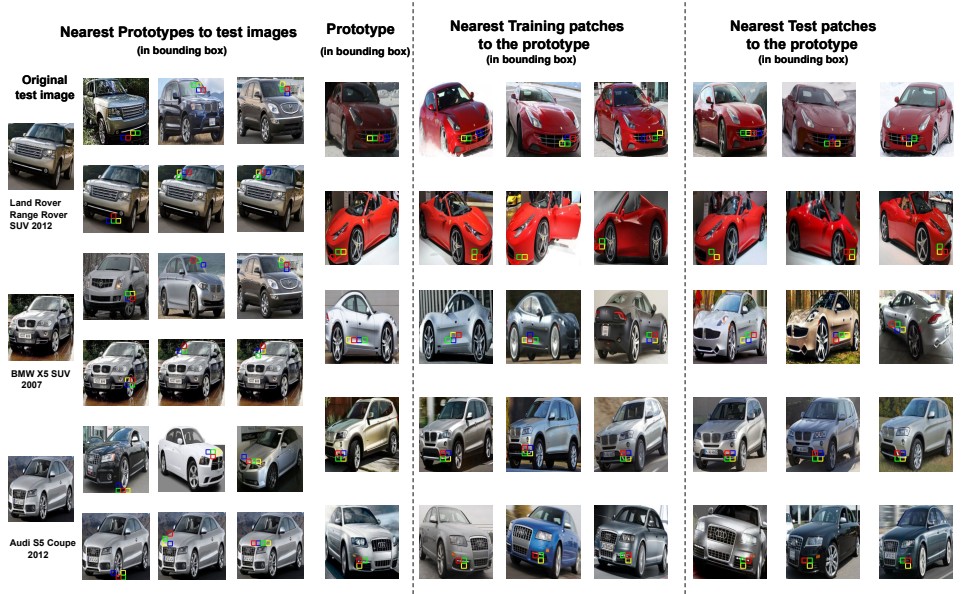

Figure 38: Examples of local analysis (left) and global analysis (right) of ProtoViT with Deit-tiny backbone on the Stanford Cars dataset.

be used for important computer vision applications to discover new knowledge and create better human-AI interfaces for difficult, high-impact applications.

## N  Computational Cost

By introducing greedy matching and coherence loss, we do not observe a significant increase in the computational cost. On the other hand, the computation of the adjacency mask may rely on the power of CPUs, as it involves more of matrix broadcasting and iterations. On a 13th Gen Intel R Core(TM) i9-13900KF CPU, it takes 0.2 seconds to compute each iteration with batch size 128, and $r = 1$ with 2000 prototypes. Overall, it takes roughly 16 hours to train the model with DeiT-Small backbone on the bird dataset, which is a similar amount of training time as ProtoPool [15] and TesNet [46] using a similar amount of backbone parameters.

## O  Training software and platform

We implemented our ProtoViT using Pytorch. The experiments were run on 1 NVIDIA Quadro RTX 6000 (24 GB), 1 NVIDIA Ge Force RTX 4090 (24 GB) or 1 NVIDIA RTX A6000 (48 GB).

