# OpenReview forum: "Interpretable Image Classification with Adaptive Prototype-based Vision Transformers"
_NeurIPS.cc/2024/Conference — NeurIPS 2024 poster_

### Official Review · Reviewer_Mc2e · 2024-07-03

**Soundness:** 2
**Presentation:** 3
**Contribution:** 2
**Rating:** 6
**Confidence:** 3

**Summary:**

The authors present a novel method for interpretable image classification by incorporating a vision transformer (ViT) into the prototypical neural network framework which provides case-based reasoning to neural network based image classifiers. They claim that most existing prototypical methods are convolutional neural network (CNN)-based, and those methods are limited by spatially rigid prototypes, thus failing to handle geometric variations of objects. Existing methods that try to handle this geometric variation either rely on a continuous latent space, which is not compatible with ViTs, or are other prototype-based ViT that fail to provide inherently interpretable explanations. Due to these existing problems, the author present ProtoViT with the following contributions:
Incorporates a ViT backbone that can adaptively learn interpretable prototypes that can handle geometric variation of different sizes.
They achieve the above with a greedy matching algorithm utilizing an adjacency mask and an adaptive slots mechanism.
They give empirical evaluation showing SOTA accuracy and a qualitative analysis showing the faithfulness and coherence of the learned prototype representations.

**Strengths:**

Soundness:

The methods are clean and sound with ample ablation experiments, and it appears their approach can perform better than others (marginally) and have interpretable and coherent prototypes.

Presentation:

The paper was easy to follow with clear claims, ideas and methods. While some of the figures could be a bit cleaner (such as the boundaries and borders in Figure 4), and some notation seemed a bit odd, they communicated their ideas/methods well.

Contribution:

The paper gives a clean method for utilizing ViTs in the prototypical framework of deeplearning for interpretability and even incorporates existing methods for making prototypes more flexible via utilizing the approach from Deformable ProtoPNet. They incorporate a novel coherence loss that encourages sub prototypes to be similar to each other. In addition, this paper utilizes a greedy matching algorithm with an adaptive mask to learn geometrically local sub-prototypes. This method also allows for an adaptive number of sub-prototypes through the slot pruning mechanism.

**Weaknesses:**

Soundness:

The lack of qualitative comparison with other ViT methods. I know the authors state that these other vision transformer methods do not project the learned prototypical features to the closest latent patches, but they still provide explanations. Could more be expanded on this and/or a figure showing this lack of reasoning/inherent interpretability?
- This is my biggest concern

Presentation:

In the section 3.4 for “Optimization of last layers” did you mean “... l-th class prototypes…” with a plural on the prototypes? This was unclear to me

Contribution:

However due to already preexisting ViT prototype methods (with a lack of comparison to them), the contribution this ViT makes compared to others is unclear.

**Questions:**

Examples in the paper only show good outcomes. What do incorrect predictions look like, and does the explanation (prototypes) show a reason for why the model was incorrect?

Do the authors have additional comments about potential information leakage between image patches in the attention step occurring in the activation of the prototypes? I ask this in context of a latent patch containing more information regarding other patches around it contributing to the learned prototype. When we project on this prototype, we look at its spatial position and project to the real image, but it may not tell the whole story.

When determining the next sub-prototype, why do you only consider the last sub-prototype when creating the adaptive mask given radius r? Why not all the currently selected sub-prototypes? Wouldn’t you get more coverage and cohesive information if considering them all?

Why can’t prototypes be shared across classes? While I still think the prototypes learned are good, I think a limitation that wasn’t stated is the lack of across class prototype sharing.

I feel like the coherence loss could limit your overall prototype representation. If all sub-prototypes have to be similar, then wouldn’t that hurt a prototype representation that has distinct parts. I’m thinking of a potential example of three differently colored stripes being next to each other. If the sub-prototypes are each of different colors, wouldn’t they be very different; thus discarded due to this loss? This is even shown in E.3. I know that is where the adaptive mask could combat this problem, but I would like more discussion on this.

I’m curious how sensitive your method is to masking / perturbation. For example, if you mask out some or all the sub-prototypes for the top activated area in the test image, can the model still find the prototype elsewhere if it still exists in the image? A particular case I’m interested in is if a prototype is about the blue on a bird’s feathers, so you make that part that the model initially detected. Does the model still pick up the blue elsewhere?

Could you expand more on why you “believe that simply adding prototype layers to an architecture without well defined “cases” does not make the new architecture more interpretable.” In particular, what do you mean by ‘well defined cases’?

**Limitations:**

The method lacks across class prototype sharing.

The coherence loss lack discussion on its limitation of making sub-prototypes being similar thus hindering diverse representation which may be important in a prototype.

The paper lacks comparison to other interpretable ViT methods

---

> ### Author Rebuttal · Authors · 2024-08-07
>
> Thank you for your thorough review and comments. We are happy that you find our method sound and our experiments ample. We address your comments below:
>
> > The lack of qualitative comparison with other ViT methods..
>
> We attempted to compare our visualizations to those from ProtoPFormer [1], but are unable to show any reasoning from ProtoPFormer, since they do not have explicit prototypes and their code does not include functionality to perform analysis on their prototypes. From their code, we were only able to extract prototype-wise activation patterns on a given test image and the masked out version of the test image from the global branch. We could not find any way to visualize the prototypes yielding these activation maps, which is likely because the prototypes are not projected.
> We also attempted to visualize ViT-Net [2], but found that they have not provided any visualization code. We reached out to the authors requesting such code, but have not yet received any response. We are happy to discuss this point further during the discussion period if desired. We would like to emphasize that the other prototype-based ViTs are not able to address geometric variations. Those methods incorporate non-deformable CNN layers after the ViT encoder. Unfortunately, those methods do not project prototypes to the closest latent patches and thus have no visualizations. Moreover, even if they had a visualization of their prototype, it would be similar to ProtoPNet which we directly compared in Fig. 1. In short, we agree that such visual comparisons would be useful, but as discussed above, this is unfortunately not possible. We will however add parts of the discussion above to our revised paper. We hope this addresses your concerns.
>
> > In the section 3.4 for “Optimization..
>
> We agree that the current phrasing is confusing – we will change it to  “... associated with class b for each prototype from class l”.
>
> > wrong prediction examples
>
> Thank you for the great suggestion. We have added some examples of incorrect reasoning to the shared response. Interestingly, we found that some of the time where the algorithm appears to have gotten it wrong, it was actually because the dataset was mislabeled and the algorithm was actually correct. We present examples of this in the shared response and will add them to our revised paper.
>
> > ..  why only consider the last sub-prototype..
>
> Considering the neighbors of only the most recently selected part helps encourage prototypical parts to have a consistent relationship – i.e., the third part must always be adjacent to the second. If a prototype represents a bird’s ankle, it should always be true that the leg part is adjacent to the foot part, which is adjacent to the toe part.
>
> Moreover, we have found that further restricting where sub-prototypes activate improved the semantic consistency of prototypes. The extreme case of this – removing adjacency masking entirely – is shown in Appendix Fig 8, where a prototype without any masking confuses a bird’s beak with another’s feet. As such, we aimed to keep adjacency masks tight by selecting only neighboring cells to the most recently activated sub-prototype.
>
> > Information Leakage:
>
> This is another great question. Due to character limit, please refer to the global response and the response to reviewer **GC9W**. We have used empirical evaluation to show that the theoretical possibility of information leakage isn’t happening to our model.
>
> > shared across classes?
>
> Prototypes could indeed be shared across classes if desired – ProtoViT is compatible with methods like ProtoPool [3] and ProtoPshare [4]. For the purposes of this work, we used a simple linear layer between prototype activations and class predictions for ease of training. We wanted to highlight the impact of the novel elements of this work without combining too many features from the literature, in order to keep our contribution clear.
>
> Thanks again for the question, we will clarify this in our revised paper.
>
> > the coherence loss could limit overall prototype representation.
>
> Coherence loss only encourages the sub-prototypes to be semantically similar so that each prototype will represent only one semantic concept. This makes prototypes simpler to understand, since they represent a single, intuitive concept.
> It is important to note that this coherence is enforced **in the latent space**. Every part of a sub-prototype does not need to be visually identical; they simply have to be semantically similar. If the three stripes described are really important for identifying the species, we might expect them to be encoded close to each other in the latent space, and thus be allowed to be grouped together by coherence loss. Finally, if so desired by the user, the coherence loss can be de-emphasized through tuning its corresponding hyperparameter. The inclusion of the coherence loss allows the users to control this aspect of the prototypes as in Appendix E.3.
>
>
> > perturbation?
>
> This is a great question. We have included examples in the shared pdf and discuss this in our global response.
>
> > ‘well defined cases’?
>
> This is a very key point that we are happy to further clarify. By well-defined cases, we mean that every prototype should be explicitly tied to one or more images so that we can visualize them. In both ProtoViT and the original ProtoPNet, this is achieved by projecting each prototype to be exactly equal to part of the latent representation of some training image. We can then refer back to the training image for a visual representation of the prototype. Without such a mechanism to enable visualizations, prototypes are just arbitrary learned uninterpretable tensors in the latent space of the network, which are not that different from a convolutional filter in any black box model.
>
> We again thank the reviewer for their thoughtful comments. We hope we have addressed all your concerns in our response and are happy to provide additional clarifications if needed.

---

> ### Author Response · Authors · 2024-08-07
> **citations**
>
> [1] Xue, Mengqi, et al. "Protopformer: Concentrating on prototypical parts in vision transformers for interpretable image recognition." arXiv preprint arXiv:2208.10431 (2022)
>
> [2] Kim, Sangwon, Jaeyeal Nam, and Byoung Chul Ko. "Vit-net: Interpretable vision transformers with neural tree decoder." International conference on machine learning. PMLR, 2022.
>
> [3] Rymarczyk, Dawid, et al. "Interpretable image classification with differentiable prototypes assignment." European Conference on Computer Vision. Cham: Springer Nature Switzerland, 2022.
>
> [4] Rymarczyk, Dawid, et al. "Protopshare: Prototype sharing for interpretable image classification and similarity discovery." arXiv preprint arXiv:2011.14340 (2020).

---

> ### Comment · Reviewer_Mc2e · 2024-08-11
> **Rebuttal Response**
>
> Thank you for a detailed rebuttal!
>
> I understand that other methods may not project to a latent of a training example, but this doesn't seem like a big step / modification to add to the other methods for comparison. If this was done then having this comparison would have added strength to this paper. I understand that they would just be similar to ProtoPNet if this was added, but they wouldn't be the same. Thank you for emphasizing that other ViT-based methods are unable to handle geometric variations. Thank you for adding parts of this discussion in the paper if accepted. On this point overall, having a unifying comparison between these approaches would have contributed well to this area.
>
> Thank you for the perturbation and misclassification examples! Please do add this to the final paper if accepted!
>
> I appreciate the response on the coherence loss. I think further analysis of this loss and it's limitations would be desirable. I understand that this is in the latent space, but the concern is still that same. If several parts are semantically different, but their combination is essential for the downstream task, then I still think this loss would restrict that. I understand this loss adds more control which is a reason I added it as a strength, but I still think it comes with limitations (unless there are more analysis that says otherwise).
>
> Thank you for addressing some of my other concerns.
>
> I've updated my score to reflect the added strength from the rebuttal.

---

> ### Author Response · Authors · 2024-08-11
>
> Thank you for your response!
>
> > I understand that other methods may not project to a latent of a training example, but this doesn't seem like a big step / modification to add to the other methods for comparison. If this was done then having this comparison would have added strength to this paper.
>
> We agree that adding visualizations from other VIT based models would be very interesting and could directly show the strength of our method. However, since those methods also learn prototypes on the class tokens which have no correspondence to the image as we discussed in the related work, it is unclear how to project and visualize those prototypes even with projection. The main reason that those methods do not perform projection is because they observe a dramatic performance drop. This is explicitly stated on page 11 of ProtoPFormer, where they say:
>
> "Our proposed ProtoPFormer does not employ the “push” process for two main reasons. One is that this process causes the performance degradation with ViT backbones, and the other is that our global and local prototypes are the high-level abstraction of associated visual explanations, representing their features based on the whole training set."
>
> This drop is because that their method is not able to learn a well clustered latent space, which reviewer **GK9W** agreed on. Thus, it is extremely unclear how those methods produce the visualizations used in their manuscripts.
>
>
> >Thank you for the perturbation and misclassification examples! Please do add this to the final paper if accepted!
>
> We will make sure to do so!
>
> >If several parts are semantically different, but their combination is essential for the downstream task, then I still think this loss would restrict that.
>
> When they are semantically different, it would be captured by other prototypes associated with that class as encouraged by orthogonality loss, which is why we adopt the orthogonality loss in our method. Coherence loss is encouraging the similarity inside a prototype (which consists of some sub-prototypes), and **orthogonality loss is encouraging prototypes to be different from each other (to have diverse representation)**. So, if there are several semantically different parts that are each necessary for classification, a different prototype should learn to identify each one of them. We will make sure to discuss this more in the final manuscript.
>
> We hope this would further address your concerns. And thank you again for updating your score and engaging with us! We greatly appreciate your feedback.

---

### Official Review · Reviewer_GK9W · 2024-07-08

**Soundness:** 2
**Presentation:** 3
**Contribution:** 3
**Rating:** 5
**Confidence:** 5

**Summary:**

The authors introduce ProtoViT, a model that leverages the Visual Transformer (ViT) architecture and integrates prototypical parts for case-based reasoning. This method is self-explainable, adhering to the rule "this looks like that." A novel aspect of ProtoViT is the use of prototypical parts of varying sizes, utilizing a ViT backbone. The authors claim that these prototypical parts are coherent and inherently interpretable. They evaluate ProtoViT on the CUB and Stanford Cars datasets, using accuracy as the metric for comparison. The training process for the model involves five loss components in addition to cross-entropy.

**Strengths:**

The paper is well-written, with images effectively illustrating the intended concepts. The introduction of the greedy matching algorithm is particularly engaging and holds significant importance for the community. The introduction section is well-crafted, clearly outlining the contributions. Additionally, the computational experiments are thorough and demonstrate comprehensive accuracy.

**Weaknesses:**

The primary concern lies in ensuring that the ViT backbone can maintain prototypical parts that are both local and inherently interpretable. Since ViT uses attention mechanisms that mix information from all patches, there is a risk of confusion for the end user. To address this, I suggest conducting a spatial misalignment benchmark [1] to analyze its influence.

Additionally, there are no metrics related to explainability demonstrating whether the model improves interpretability, such as with FunnyBirds [2] or through a user study [3].

[1] Sacha, Mikołaj, et al. "Interpretability benchmark for evaluating spatial misalignment of prototypical parts explanations." Proceedings of the AAAI Conference on Artificial Intelligence. Vol. 38. No. 19. 2024.
[2] Hesse, Robin, Simone Schaub-Meyer, and Stefan Roth. "FunnyBirds: A synthetic vision dataset for a part-based analysis of explainable AI methods." Proceedings of the IEEE/CVF International Conference on Computer Vision. 2023.
[3] Kim, Sunnie SY, et al. "HIVE: Evaluating the human interpretability of visual explanations." European Conference on Computer Vision. Cham: Springer Nature Switzerland, 2022.

**Questions:**

How are you measuring all the claims, especially coherence and inherently interpretable models?
Could you provide a prototype purity metric for PIP-Net?
Is it possible to analyze whether ViT can be a backbone for prototypical parts or not, test with spatial misalignment benchmark?

**Limitations:**

There is no quantification of interpretability, no user-study, nor no reference to XAI benchmarks such as FunnyBirds and spatial misalignment benchmarks.

---

> ### Author Rebuttal · Authors · 2024-08-07
>
> Thank you for your thorough review and comments. We are happy that you find our matching algorithm engaging and a significant contribution and our experiments thorough and comprehensive. We address your comments below:
>
> > The primary concern lies in ensuring that the ViT backbone can maintain prototypical parts that are both local and inherently interpretable. Since ViT uses attention mechanisms that mix information from all patches, there is a risk of confusion for the end user. To address this, I suggest conducting a spatial misalignment benchmark [1] to analyze its influence.
>
> Thank you for the great suggestion. As suggested, we have run experiments on the benchmark [1], and found our model outperforms models such as ProtoPool [2] and ProtoTree [3] that are CNN based architectures. We present the complete results in our shared response. Our model achieves very close performance in PLC to ProtoPNet [4] and strictly better performance in PAC and PRC. Our model is also robust to adversarial attacks with roughly only 2 - 3% drop in accuracy. These results suggest that our model with ViT can create prototypical parts that are both local and inherently interpretable equally or better than many of the influential existing models with CNN backbones. We will add these results to our revised paper.
>
> Further, in section G of the appendix, we present several examples of global analysis. These figures show the highest activations of a variety of prototypes on training and test images, and consistently show that each prototype activates highly on a single semantic concept. This consistency suggests that our visualizations are **a good representation of the prototypes**, and not just spuriously selecting nice patches.
>
> > Additionally, there are no metrics related to explainability demonstrating whether the model improves interpretability, such as with FunnyBirds [2] or through a user study [3].
>
> We apologize for the confusion, we did indeed run a user study based on the agreement task from HIVE, which is your reference [3]. It’s in Appendix I, labeled in the table of contents in the appendix as a user study. The study shows that our model improves interpretability, and shows statistically significantly improvements in user understanding and confidence on the model reasoning process. We will make sure the user study is clearly discussed in the main body of the revised paper.
>
> > How are you measuring all the claims, especially coherence and inherently interpretable models? Could you provide a prototype purity metric for PIP-Net? Is it possible to analyze whether ViT can be a backbone for prototypical parts or not, test with spatial misalignment benchmark?
>
> We agree that these are important clarifications. For coherence, we have a definition in the training algorithm, Section 3.4, Equation 4. “Inherently interpretable models” are constrained to make their reasoning processes easier to understand, see work [5]. Our paper’s prototype-based reasoning is a constraint on the network to make its reasoning process easier to understand. The result from the newly run misalignment benchmark described above and in our global response shows that ViT can work as good as CNN backbones in our proposed method.
>
> Finally, while we agree that a purity metric would be useful, however, given the limited time to prepare our rebuttal, we decided to focus on other experiments, such as the misalignment experiments on the new benchmark [1] suggested by the reviewer (which we deemed more important). In our understanding, the purity metric simply quantifies whether prototypes consistently activate on the same concept, which we show to be the case through many examples of global analysis. We expect the purity metric to be strong based on our global and local analysis results and we are working on computing the purity metric to add to our revised paper.
>
> Thanks again for the thorough review and the great suggestions. We believe we have addressed the main concerns raised by the reviewer. We are happy to provide additional clarifications if needed.
>
> [1] Sacha, Mikołaj, et al. "Interpretability benchmark for evaluating spatial misalignment of prototypical parts explanations." Proceedings of the AAAI Conference on Artificial Intelligence. Vol. 38. No. 19. 2024.
>
> [2] Rymarczyk, Dawid, et al. "Interpretable image classification with differentiable prototypes assignment." European Conference on Computer Vision. Cham: Springer Nature Switzerland, 2022.
>
> [3] Nauta, Meike, Ron Van Bree, and Christin Seifert. "Neural prototype trees for interpretable fine-grained image recognition." Proceedings of the IEEE/CVF conference on computer vision and pattern recognition. 2021.
>
> [4] Chen, Chaofan, et al. "This looks like that: deep learning for interpretable image recognition." Advances in neural information processing systems 32 (2019).
>
> [5] Rudin, Cynthia, et al. "Interpretable machine learning: Fundamental principles and 10 grand challenges." Statistic Surveys 16 (2022): 1-85.

---

> > ### Comment · Reviewer_GK9W · 2024-08-09
> >
> > That is indeed a great response to my review. Thank you for the clarifications and discussion. If your work is accepted, please include values for the purity metric, as it is an established metric for evaluation. Including this will enhance the consistency of the research.
> >
> > The results of the spatial misalignment benchmark are surprising, as I did not expect the ViT to be more robust in terms of explanation quality and faithfulness. Good job!
> >
> > Regarding the user study, I really appreciate the presentation of results from Ma et al. [1], who also tested if users performed better than random. It would be worthwhile to check this for your method as well.
> >
> > After consideration, I am leaning toward increasing my grade, but I still need to digest the information from other reviewers and their discussions.
> >
> > [1] Ma, Chiyu, et al. "This looks like those: Illuminating prototypical concepts using multiple visualizations." Advances in Neural Information Processing Systems 36 (2024).

---

> > > ### Author Response · Authors · 2024-08-09
> > >
> > > Thank you for your response!
> > >
> > > >  If your work is accepted, please include values for the purity metric, as it is an established metric for evaluation. Including this will enhance the consistency of the research.
> > >
> > > We will make sure to include the purity metric to the final manuscript if the paper is accepted.
> > >
> > > > Regarding the user study, I really appreciate the presentation of results from Ma et al. [1], who also tested if users performed better than random. It would be worthwhile to check this for your method as well.
> > >
> > > We agree that this could be an interesting addition to our paper, but it will take a long time to design and complete such a user study. We don't believe that we can finish it before the end of the discussion period. We will add it to the final manuscript if the paper is accepted.
> > >
> > > Thanks again for engaging with us!

---

### Official Review · Reviewer_w1UG · 2024-07-11

**Soundness:** 4
**Presentation:** 3
**Contribution:** 4
**Rating:** 7
**Confidence:** 5

**Summary:**

This paper presents a novel strategy to learn interpretable visual prototypes for visual transformers, with a good property of offering spatially deformed prototypes. The method also introduce an slot mechanism which can learn an adaptive number of prototypical parts. The proposed are wisely designed for visual transformer architectures.

**Strengths:**

1. It is nice to draw inspiration from the focal similarity when computing the  patch features.
2. Different from Deformable ProtoPNet, the proposed method present a new way to accommodate geometric variations of objects.
3. The proposed method is validated on two benchmarks and with extensive ablation studies.
4. In general, the paper is well written.

**Weaknesses:**

1. The first concerns is about the use of deformable prototypes that containing K sub-prototypes, which means the proposed method will have more learnable prototype vectors (K times) than ProtoPNet, TesNet, ProtoPFormer, and so on. These previous approaches only use 1*1 prototypes. Does the performance improvement of the proposed method come from the largely increased number of sub-prototypes?
2. The idea of greedy matching algorithm is similar to the greedy prototype projection, proposed in [1, 2]. The authors are suggested to state their difference of the related works. Also, some important work using prototypes for interpretable image classification should be reviewed and discussed, such as [3, 4, 5].
3. Regarding the adaptive slots mechanism, it is good for the motivation of to learn an additional indicator to measure the importance of sub-prototypes. From my understanding, the learnable vector v is like a gate, which should be saved as model parameters after training. One potential limitation is such mechanism introduces an extra gate parameter v, compared with previous methods.
4. It not much clear about the adjacency masking. Since the prototypes are not initialized to have the position information, how do choose the patch/feature tokens around the prototypes within r?
5. The authors mention the issue of performance degradation after prototype projection. Does the proposed method still suffer from such issue? What extent will the performance drop?
6. The method has too much loss coefficients, which are selected without detailed tuning procedure.


References:

[1] Knowledge Distillation to Ensemble Global and Interpretable Prototype-Based Mammogram Classification Models

[2] Pixel-grounded prototypical part networks

[3] PIP-Net: Patch-Based Intuitive Prototypes for Interpretable Image Classification

[4] Learning support and trivial prototypes for interpretable image classification

[5] Concept-level debugging of part-prototype network

**Questions:**

N/A

---

> ### Author Rebuttal · Authors · 2024-08-07
>
> Thank you for your thorough review and comments. We are happy that you find our method novel and well-designed for transformers. We address your comments below:
> > The first concerns is about the use of deformable prototypes that containing K sub-prototypes, which means the proposed method will have more learnable prototype vectors (K times) than ProtoPNet, TesNet, ProtoPFormer, and so on. These previous approaches only use 1*1 prototypes. Does the performance improvement of the proposed method come from the largely increased number of sub-prototypes?
>
> Thanks for the important question. We discuss this point in section F of the appendix. While our prototypes do consist of more parts than prior work, they represent a similar proportion of each input, since the latent space of our backbones have a finer spatial resolution. We are not sacrificing locality, or changing the general reasoning process. We only add flexibility to the prototypes by decomposing them into multiple, flexible parts. In fact, when using 1*1 prototypes, the model performance is **85.28 +/- 0.11** which is comparable to our final model. So, the improved performance is not necessarily because of the increased number of sub-prototypes.
>
> However, **the interpretability of the earlier method is very limited because the 1*1 patches are too small to convey much information**, which is exactly where the motivation of our work comes from. We are open to moving this section to the main body for the camera ready if the reviewer thinks that is warranted.
>
> >  Regarding the adaptive slots mechanism, it is good for the motivation of to learn an additional indicator to measure the importance of sub-prototypes. From my understanding, the learnable vector v is like a gate, which should be saved as model parameters after training. One potential limitation is such mechanism introduces an extra gate parameter v, compared with previous methods.
>
> The reviewer is correct in that there is an additional parameter associated with each part of each prototype (the slot indicator), but we argue that this is not a limitation - it’s a **benefit**. This indicator allows us to learn prototypes with different numbers of parts, increasing the **flexibility** of the network and helping us reduce the overall number of prototypical parts when possible. Like adding any parameter to any network, there are benefits and drawbacks, but we confirm empirically that the benefits outweigh the drawbacks for performance. And because of this, we can learn prototypes that capture more specific features such as the red eyes in Fig. 4. Without these additional parameters, instead of learning the exact red eye feature, it will only learn the head of the bird as a whole. This is an important discussion which we will include in our revisions.
>
> > It not much clear about the adjacency masking. Since the prototypes are not initialized to have the position information, how do choose the patch/feature tokens around the prototypes within r?
>
> We apologize for the confusion. Adjacency masking is tied in with the greedy matching procedure: we match the first part of a prototype to anything in a given image, mask out every latent patch more than radius r away from the selected patch, and greedily select the best match for the next part from among the cells that were not masked out. We hope this clarification helps.
>
> >  The authors mention the issue of performance degradation after prototype projection. Does the proposed method still suffer from such issue? What extent will the performance drop?
>
> Yes, like other prototype based methods, we do see a small drop in performance after projection relative to before. The drop in performance is usually around 0.8 to 1% accuracy, but we report only accuracy values taken following the projection step. That means the strong performance we report already accounts for this slight drop in performance.
>
> As described in Theorem 2.1 from ProtoPNet [1], under a perfect training setting, the drop in performance should be negligible. The big drop in performance typically seen is mainly caused by the fact that the prototypes are not trained semantically close enough to the latent patches. Since the prototypes trained in our method are semantically much closer to the latent patches, our drop in performance is minimal, as discussed above.
> We will make this clear in our revised paper.
>
> > The method has too much loss coefficients, which are selected without detailed tuning procedure.
>
> Like any hyperparameter, these loss coefficients can be tuned by a variety of techniques. For simplicity, we used the values suggested in prior work for terms that already existed, and used a small grid search to select the other values. No dedicated tuning procedure is necessary for these coefficients. Additionally, we note that the number of coefficients in our loss (five) is comparable to other similar work, such as TesNet [2]. We will clarify this in the revisions.
>
> [1] Chen, Chaofan, et al. "This looks like that: deep learning for interpretable image recognition." Advances in neural information processing systems 32 (2019).
>
> [2] Wang, Jiaqi, et al. "Interpretable image recognition by constructing transparent embedding space." Proceedings of the IEEE/CVF international conference on computer vision. 2021.

---

> > ### Comment · Reviewer_w1UG · 2024-08-11
> >
> > Thank you for the responses. I am happy with your work.
> > It is great to have an exhaustive discussion about the performance drop caused by prototype projection. I hope such discussion can be included in the main paper, if your work is accepted, since a few prior work noticed this practical issue. Good job!

---

> > > ### Author Response · Authors · 2024-08-11
> > >
> > > Thank you very much for your response! We will make sure to include the discussion of the performance drop caused by projection in the body of the final manuscript. It will be _discussed in the training algorithm section after the discussion of projection starting on page 7_. It is encouraging to know that our detailed revisions have addressed your concerns. Your recognition of our work in terms of contribution, novelty, soundness, and presentation is greatly appreciated.
> > >
> > > We also appreciate it _if you could consider updating the rating_, as this would kindly reflect your most recent evaluation and kind recognition of the paper and acknowledge our efforts.
> > >
> > > Thank you again for engaging with us!

---

### Official Review · Reviewer_dybc · 2024-07-24

**Soundness:** 3
**Presentation:** 3
**Contribution:** 2
**Rating:** 6
**Confidence:** 4

**Summary:**

The paper presents ProtoViT, a method for interpretable image classification. ProtoViT incorporates ViT backbone with deformed prototypes that explains its predictions. ProtoViT consists of three components:

(1) a feature encoding layer with a pre-trained ViT backbone, which computes a latent representation of an image;

(2) a greedy matching layer which compares the latent representation to learned prototypes; and

(3) an evidence layer which aggregates prototype similarity scores into a classification using a fully connected layer.

Quantitatively, ProtoViT achieves better performance than previous prototype-based methods. Qualitatively, it identifies meaningful prototypes to explain the prediction.

**Strengths:**

1. The paper is well-written and illustrates the technical details. The qualitative visualization are helpful in understanding the method. Attached appendix provides a lot of meaningful details.


2. The overall scheme of interpretable image classification is nicely presented with small patches!

**Weaknesses:**

1. It is understandable that one goal of interpretable image classification is to tell why something works. However, a bigger goal is to understand why something didn't really work. Is it possible to highlight the examples where the method could not predict the correct class? It would be then interesting to understand the reasons for failure.

2. The method relies on a strong assumption of a solid backbone model that can already achieve a good performance on the given task. This restricts the applicability of the method to limited scenarios where we do not need interpretable image classification at the first place. In such a scenarios, the nearest neighbors obtained using latent feature representation and pixel correspondences may be sufficient.

3. Limitations of the method is not clear from the paper. I could guess at certain places in the method section where things could go wrong. It would be better if the authors could illustrate those points with suitable qualitative analysis.

4. User study is not properly designed and is not statistically significant.

Overall: The paper is interesting. In a first reading, everything looks good. But then you start asking yourself questions about the different scenarios where the method will not work (and there are definitely such scenarios as can be judged from quantitative evaluation and user study) and why it will not work. The paper falls short in explaining them with details.

**Questions:**

Please see Weaknesses Section.

**Limitations:**

Authors did provide limitations but a far-fetched one that does not really talk about the limitations of the current method.

---

> ### Author Rebuttal · Authors · 2024-08-07
>
> Thank you for your comments. We are glad that you find our paper interesting, well-written and nicely presented. We address your points below:
> > ... Is it possible to highlight the examples where the method could not predict the correct class? It would be then interesting to understand the reasons for failure.
>
> Yes! Per your suggestion, we present examples of this in the shared pdf response. Interestingly, we found that some of the time where the algorithm appears to have gotten it wrong, it was actually because the dataset was mislabeled and the algorithm was actually correct (see the first example in Figure 1 of the pdf rebuttal). Appendix Fig. 8 also includes an example when the model misclassified Black Tern and Pigeon Guillemot. And this misclassification is consistent across different model algorithms because of the red feet that both birds have. We will include these additional examples in the revised paper.
>
> > The method relies on a strong assumption of a solid backbone model that can already achieve a good performance on the given task. This restricts the applicability of the method to limited scenarios where we do not need interpretable image classification at the first place. In such a scenarios, the nearest neighbors obtained using latent feature representation and pixel correspondences may be sufficient.
>
> This is an important question that needs to be clarified. Nearest neighbors is not actually sufficient. First, calculating nearest neighbors for a full dataset is extremely expensive, which means training is extremely difficult since you’d have to calculate nearest neighbors at each iteration (or at least at many iterations); this makes it impractical for any complicated task.
>
> Second, nearest neighbors are much more sensitive and difficult to troubleshoot. With prototypes, a person can look at all the prototypes and audit them, whereas you can’t do that with nearest neighbors for all the points. As we mentioned in our response to your first question, we found points that are mislabeled - if those were learned prototypes, we would have pruned them or relabeled them. We wouldn’t be able to check all those examples with nearest neighbors.
>
> Third,  we argue that the method is not limited to scenarios where we don’t need
>  interpretability. A good example is  IAIA-BL, which uses prototypes for analyzing breast lesions in mammograms [1]. We **definitely** need interpretability here, as the stakes are high. Additionally, even when a black box model performs well, it doesn’t necessarily mean that it learns well. The example in our ablation section Appendix. Fig. 8, shows a version of our model that has a higher performance than the original; however, it is clear that the model confused the beak in the test image with the feet in the prototypes.
>
> For applications that are high stakes, having an interpretable network allows us to troubleshoot the data and the model so that we can improve accuracy **above and beyond** the black box. The black box is just a good starting point for training our interpretable networks. In fact, we show that our network already has better performance than the black boxes, including the ViT-based and CNN-based models (see Table 2 of the main paper).
>
> > Limitations of the method is not clear from the paper...
>
> In addition to what is discussed in the paper, the following limitations could apply to our method:
>
> 1. Even though our method is comparable with other prototype models in terms of training speed, we should note that all prototype models take additional time compared to their black box backbone.
>
> 2. Location misalignment is also a common problem in CNN-based models. As we discuss in our global response, this issue is still present in our model, but to a much lesser extent.
> We will add these to the revised paper.
>
> > User study is not properly designed and is not statistically significant.
>
> We apologize for the confusion. The user study results are actually significant at the alpha=0.05 level (all our p-values for the t-tests are below 0.05), as shown in Table 8 of the appendix.
>
> As for the design of the user study, we followed a similar structure of the agreement task to that of the HIVE [2] paper, which is known for the quality of its human-studies experiments. However, we would be happy to consider suggestions to improve our user study.
>
> > Overall: The paper is interesting. In a first reading, everything looks good. But then you start asking yourself questions about the different scenarios where the method will not work ... The paper falls short in explaining them with details.
>
> We thank the reviewer for their kind words regarding our paper. We hope that with additional clarification to the manuscript based on your and other reviewers’ comments, we have provided more detail about our method and its benefits and potentially downsides.
>
> We believe that our method makes it easier to audit and understand models and that the prototype-reasoning gives us information on how to solve the problem, whether it is by augmenting the data to learn a specific concept. Importantly, it also tells us when we should not trust the prediction, as demonstrated above. Through comprehensive experiments and ablations, we have shown our method to perform reliably well at this task and is a strict improvement over other prototype networks as its localized parts are totally faithful to the reasoning process (since we do not require upsampling for our visualizations, something that other prototype based methods require).
> We thank you for your comments and hope we have addressed them all. Please let us know if there is anything you would like us to further clarify.
>
> [1] Barnett, Alina Jade, et al. "A case-based interpretable deep learning model for classification of mass lesions in digital mammography." Nature Machine Intelligence.
>
> [2] Kim, Sunnie SY, et al. "HIVE: Evaluating the human interpretability of visual explanations." ECCV 2022.

---

> ### Author Response · Authors · 2024-08-13
>
> We hope that our response has helped explain our work's contributions and address your concerns. Please feel free to let us know if you have any further questions. We are very happy to discuss!

---

> > ### Comment · Reviewer_dybc · 2024-08-13
> > **Thanks!**
> >
> > Thank you for the rebuttal! I have raised my scores.

---

### Author Rebuttal · Authors · 2024-08-07

We thank all the reviewers for their comments. In addition to addressing all of your concerns individually, we believe the following clarifications and additional experiments may be of interest to all of you. These additional experiments further reinforce the strength of our method and will be added to our revised paper.

**(1) Location misalignment benchmark:**

To ensure that the ViT backbone can maintain prototypical parts that are both local and inherently interpretable, we performed analysis on the benchmark [1] as suggested by reviewer **GK9W**. The quantitative result can be found in the shared pdf. Overall, we found that by integrating ViT backbone with our algorithm, it achieves equal or better performance than the influential CNN based prototype models such as ProtoPNet [2], ProtoPool [3], and Prototree [4] in terms of percentage change in location (PLC), percentage change in activation (PAC), and percentage change in ranking (PRC). For these metrics, lower is better. To ensure a fair comparison, we strictly follow the procedure of performing gradient based adversarial attacks as described in the paper. However, we would like to point out that since our prototypes only contain 4 (out of 196) or less patches, the adversarial attack performed is stronger to our model than other CNN based models, as a larger proportion of the images are perturbed for our model. Results can be found in Table 1 of the shared pdf.

**(2) Perturbation Analysis:**

Additionally, we have provided several instances of the perturbation analysis suggested by reviewer **Mc2e**, in which we mask out the region selected by each prototype, as shown in Figure 2 of the shared pdf. In each row, we mask out all matched locations for a prototype (middle-left column) using a white mask on the black wings and a black mask on the red parts, and check where that prototype activates after masking (shown in the leftmost column). We then confirm that the activated region for other prototypes remains reasonable when the mask from another prototype is applied (right two columns). We observed that after removing the preferred region by each prototype, it activates on another reasonable alternative (e.g., a red belly prototype might activate on a red back as a second choice).

Moreover, we performed the analysis on the test image with a modified background, and we observe that altering the background does not substantially impact the activation location of our ViT-based prototype, with all parts still solidly focused on the bird’s breast (again please refer to Figure 2 of the pdf). This is unlike the observation made in the original Location Misalignment Benchmark paper, where a CNN-based prototype model suffers from a dramatic shift in the activation location when there is a background change. This shows that our ProtoViT has a better location alignment than the CNN-based prototype models.

**(3) Misclassification examples:**

We provide the reasoning process of how our model misclassified a test image of a summer tanager and a slaty backed gull in Figure 1 of the shared pdf. We found that the misclassification of the given summer tanager example may be because of data mislabeling in the original dataset. A summer tanager does not have a black colored wing. That test image should indeed belong to the scarlet tanager class as the model predicted. Similar mislabeling cases also happen for Red Headed Woodpecker with image ID Red_Headed_Woordpecker_0018_183455 and Red_Headed_Woordpecker_0006_183383, as we found out when randomly selecting examples to present for the paper.

These examples showcase the ability of our method to help us understand the reasoning process of the model, not just when it is right, but also when it is wrong.

**(4) Other prototype-based ViT models:**

Finally, we would like to emphasize that some models suggested by the reviewers for comparison (e.g., ProtoPFormer [5], ViT-Net [6] and etc.) do not perform any form of prototype projection, and as such do not have any explicit visualization associated with their prototypes. Without such a mechanism to enable visualizations, prototypes are just arbitrary learned tensors in the latent space of the network, which are not that different from a convolutional filter in any black box model. In contrast, our network does perform projection, allowing each prototype to be tied directly to one well defined training image. This allows us to present clear, faithful visualizations of model reasoning.

[1] Sacha, Mikołaj, et al. "Interpretability benchmark for evaluating spatial misalignment of prototypical parts explanations." Proceedings of the AAAI Conference on Artificial Intelligence. Vol. 38. No. 19. 2024.

[2] Chen, Chaofan, et al. "This looks like that: deep learning for interpretable image recognition." Advances in neural information processing systems 32 (2019).

[3] Rymarczyk, Dawid, et al. "Interpretable image classification with differentiable prototypes assignment." European Conference on Computer Vision. Cham: Springer Nature Switzerland, 2022.

[4] Nauta, Meike, Ron Van Bree, and Christin Seifert. "Neural prototype trees for interpretable fine-grained image recognition." Proceedings of the IEEE/CVF conference on computer vision and pattern recognition. 2021.

[5] Xue, Mengqi, et al. "Protopformer: Concentrating on prototypical parts in vision transformers for interpretable image recognition." arXiv preprint arXiv:2208.10431 (2022).

[6] Kim, Sangwon, Jaeyeal Nam, and Byoung Chul Ko. "Vit-net: Interpretable vision transformers with neural tree decoder." International conference on machine learning. PMLR, 2022.

---

### Decision · Program_Chairs · 2024-09-25

**Decision:**

Accept (poster)

**Comment:**

The paper proposes to integrate deformable prototypes with vision Transformers for interpretable image classification. The initial review ratings were mixed. Yet after the rebuttal and discussion, all reviewers unanimously recommend this paper. They recognize the technical contributions and applaud the presentation of the paper. The authors are encouraged to incorporate review comments and rebuttal materials into the final version.